# Fam49b dampens TCR signal strength to regulate survival of positively selected thymocytes and peripheral T cells

Chan-Su Park[1,2]*, Jian Guan[1], Peter Rhee[1], Federico Gonzalez[3], Hee-sung Lee[2], Ji-hyun Park[2], Laurent Coscoy[4], Ellen A Robey[4†], Nilabh Shastri[1†], Scheherazade Sadegh-Nasseri[1]*

[1]Department of Pathology, The Johns Hopkins University School of Medicine, Baltimore, United States; [2]Department of Pharmaceutics, College of Pharmacy, Chungbuk National University, Cheongju, Republic of Korea; [3]Department of Nutritional Sciences and Toxicology, University of California,Berkeley, Berkeley, United States; [4]Department of Molecular and Cell Biology, University of California, Berkeley, Berkeley, United States

**\*For correspondence:**
cpark@chungbuk.ac.kr (C-SuP);
ssadegh1@jh.edu (SS-N)

†Senior author

**Competing interest:** The authors declare that no competing interests exist.

**Abstract** The fate of developing T cells is determined by the strength of T cell receptor (TCR) signal they receive in the thymus. This process is finely regulated through the tuning of positive and negative regulators in thymocytes. The Family with sequence similarity 49 member B (Fam49b) protein is a newly discovered negative regulator of TCR signaling that has been shown to suppress Rac-1 activity in vitro in cultured T cell lines. However, the contribution of Fam49b to the thymic development of T cells is unknown. To investigate this important issue, we generated a novel mouse line deficient in Fam49b (Fam49b-KO). We observed that Fam49b-KO double positive (DP) thymocytes underwent excessive negative selection, whereas the positive selection stage was unaffected. Fam49b deficiency impaired the survival of single positive thymocytes and peripheral T cells. This altered development process resulted in significant reductions in CD4 and CD8 single-positive thymocytes as well as peripheral T cells. Interestingly, a large proportion of the TCRγδ+ and CD8αα+TCRαβ+ gut intraepithelial T lymphocytes were absent in Fam49b-KO mice. Our results demonstrate that Fam49b dampens thymocytes TCR signaling in order to escape negative selection during development, uncovering the function of Fam49b as a critical regulator of the selection process to ensure normal thymocyte development and peripheral T cells survival.

## Editor's evaluation

The protein called Family with sequence similarity 49 member B (in short Fam49b) is a newly discovered negative regulator of TCR signaling that suppresses Rac-1 activity in vitro. The present manuscript analyzes in a comprehensive and solid manner the role of Fam49b during thymic T cell development. It is a very valuable piece of work in that it demonstrates that the Fam49b protein dampens thymocyte TCR signaling allowing thymocytes to escape negative selection.

## Introduction

Developing T cells in the thymus follow an ordered progression from CD4-CD8- double negative (DN), to CD4+CD8+ double positive (DP), and finally to CD4 or CD8 single positive (SP) T cells (*Xu et al., 2013*). Positive selection, negative selection, and CD4/CD8 lineage fate commitment of DP thymocytes rely on the strength of the interactions between TCR and self-peptides-MHC complexes

(*Hogquist, 2001*). Inadequate interactions lead to 'death by neglect' whereas overly strong interactions lead to the elimination of thymocytes through 'negative selection.' Thus, only those T cells receiving a moderate TCR signal strength are positively selected and further develop into mature T cells (*Hogquist, 2001*; *Gascoigne et al., 2016*; *Klein et al., 2014*). The TCR signal strength is also critical for CD4/CD8 lineage commitment. Enhancing TCR signaling in developing thymocytes favors the development of the CD4 lineage, whereas reducing TCR signaling favors the development of the CD8 lineage (*Hernández-Hoyos et al., 2000*; *Kappes et al., 2005*).

While the majority of thymocytes bearing high-affinity TCR for self-peptide MHC complexes undergo negative selection, not all self-reactive thymocytes follow this rule. Instead, these subsets of self-reactive non-deleting thymocytes are diverted to alternative T cell lineages through a process known as agonist-selection (*Baldwin et al., 2004*; *Stritesky et al., 2012*). Several agonist selected T cell subsets have been defined including the CD8αα+TCRαβ+ intraepithelial lymphocytes (CD8αα+T-CRαβ+ IELs), invariant natural killer T cells (iNKT cells), and Foxp3+ Regulatory T cells (Treg cells) (*Lambolez et al., 2007*; *Kronenberg and Gapin, 2002*; *Hsieh et al., 2012*). Functionally, agonist-selected T cells are thought to have a regulatory role in the immune system.

Actin cytoskeleton dynamics are important for multiple aspects of T cell function, including TCR signaling and adhesion, migration, differentiation, and execution of effector function (*Burkhardt et al., 2008*; *Kumari et al., 2014*; *Billadeau et al., 2007*). In particular, actin cytoskeleton remodeling is required to provide scaffolding for TCR signaling proteins and for maintaining a stable immunological synapse between T cells and antigen-presenting cells (APCs) (*Kaizuka et al., 2007*; *Babich et al., 2012*; *Babich and Burkhardt, 2013*). However, the mechanisms that link actin cytoskeleton dynamics to the T cell signaling are not well understood. It has been reported that T cells cytoskeletal reorganization and regulation of actin dynamics at the immunological synapse are regulated by the Rho family of small guanosine triphosphatases (Rho-GTPases) such as Rac (*Burkhardt et al., 2008*). Most members of Rho-GTPases exist in two conformational states between inactive (GDP-bound) and active (GTP-bound) (*Tybulewicz and Henderson, 2009*). The switch between the GDP- and GTP-bound states is tightly regulated by guanine nucleotide exchange factors (GEFs) and GTPase-activating proteins (GAPs). GEFs activate Rho-GTPases by promoting the exchange of GDP for GTP, whereas GAPs inhibit Rho-GTPases by stimulating their GTP hydrolysis activity. Vav family proteins (Vav1, Vav2, and Vav3) are GEFs for Rac. Active Rac-1 transduces signals by binding to effector proteins such as PAK and WAVE2 complex. Vav, Rac, and Pak play crucial roles in T cell development. For example, studies of mice lacking Vav-1 have shown that T cell development is partially blocked at pre-TCR β selection and is strongly blocked in both positive and negative selection (*Turner et al., 1997*; *Fischer et al., 1995*; *Zhang et al., 1995*). Mice lacking both isoforms of Rac1 and Rac2 show defects in pre-TCR β-selection at DN thymocytes and positive selection of DP thymocytes (*Dumont et al., 2009*; *Guo et al., 2008*). Mice lacking Pak2 show defects in pre-TCR β-selection of DN thymocytes, positive selection of DP thymocytes, and maturation of SP thymocytes (*Phee et al., 2014*).

Fam49b has been identified as an inhibitor of TCR signaling through binding with active Rac-1/2 in Fam49b-KO Jurkat T cells (*Shang et al., 2018*). Those studies showed that lack of Fam49b led to hyperactivation of Jurkat T cells following TCR stimulation, as measured by the enhancement of CD69 induction, Rac-PAK axis signaling, and cytoskeleton reorganization (*Shang et al., 2018*). Since TCR signaling strength controls thymocyte development, we hypothesized that Fam49b would be critical for thymocyte development in vivo and investigated this using a novel knockout mouse line. Here, we demonstrate that Fam49b is dispensable for positive selection but is required for negative selection by preventing overly robust elimination of thymocytes. Moreover, Fam49b-deficient peripheral naïve T cells showed impaired survival. Thus, we report that Fam49b is a critical regulator of negative selection and peripheral T cells survival.

## Results

### Generation of Fam49b-KO mice and Fam49a-KO mice

To assess the role of Fam49b in T cell development, we generated Fam49b-KO mice by creating a premature stop codon in exon 6 of the *Cyrib* locus encoding Fam49b using CRISPR/Cas9 (*Figure 1A*). Fam49a is a homologous protein that is ~80% identical to Fam49b that has also been suggested to be involved in lymphopoiesis in zebrafish (*Li et al., 2016*). We generated Fam49a-KO mice in a similar

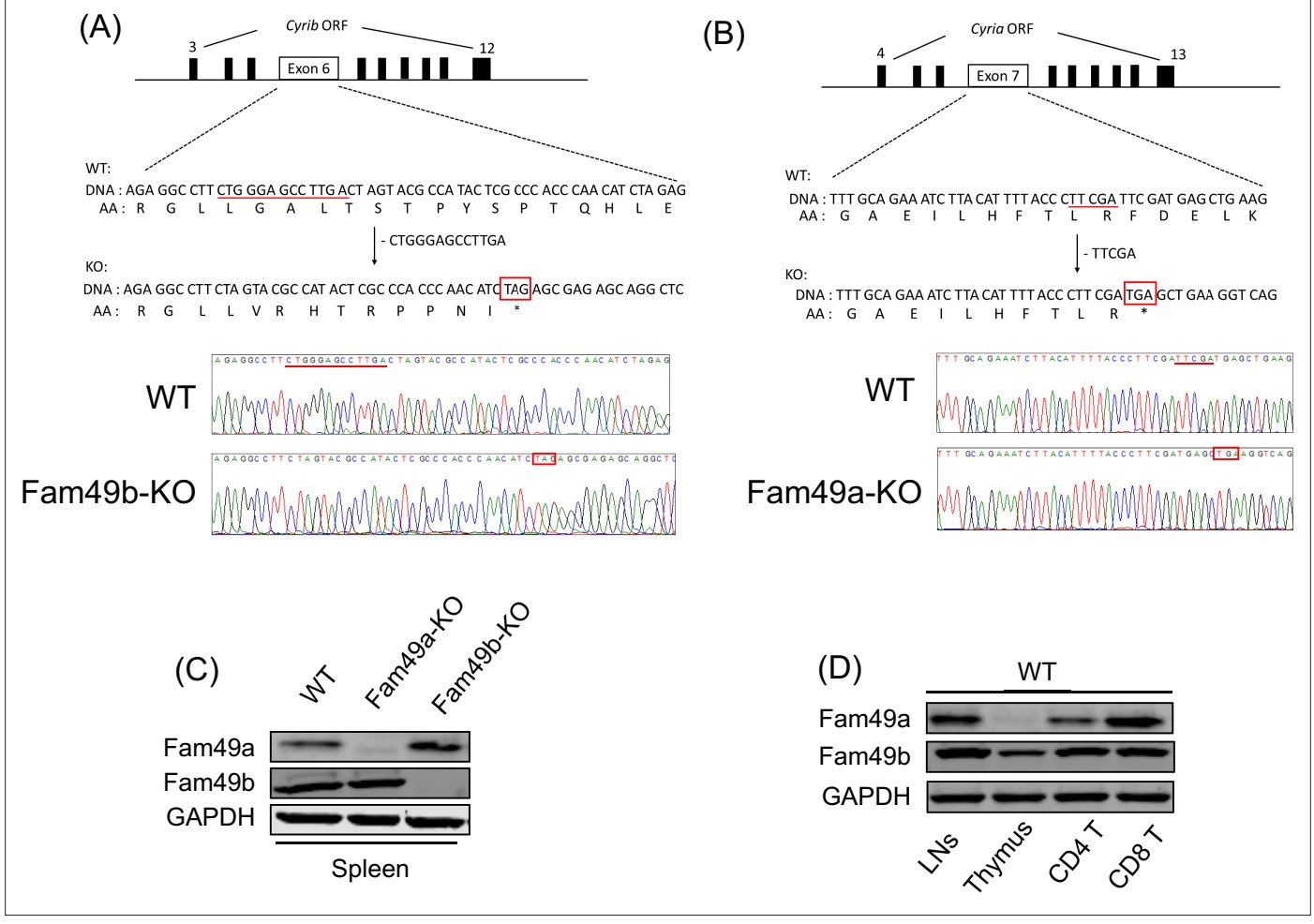

**Figure 1.** Generation of Family with sequence similarity 49 member A (Fam49a)-KO and Family with sequence similarity 49 member B (Fam49b)-KO mice with CRISPR/Cas9 and expression of Fam49a and Fam49b in mice. (**A**) Schematic diagram depicting the locations of guide RNAs (gRNAs) targeting the Fam49b (**Upper**). Representative Sanger sequencing chromatograms confirmed the genotype of Fam49b-KO mice (**Lower**). Red underline above the chromatograms indicates the deletion (5 bp) and pre-mature stop codon sequences are in red boxes. See also *Figure 1—source data 1*. (**C**) Immunoblot analysis of Fam49a and Fam49b expression in spleen from wild-type (WT), Fam49a-KO mice, and Fam49b-KO mice. Fam49a antibody (Sigma, SAB 1103179) is specific for an epitope mapping between amino acids 56–70 near the N-terminus of the human Fam49a molecule. Fam49b antibody (Santa Cruz, D-8) is specific for an epitope mapping between amino acids 8–20 near the N-terminus of human Fam49b molecule. The data are representative of three independent experiments. See also *Figure 1—source data 2*. (**D**) Immunoblot analysis of Fam49a and Fam49b expression in lymph nodes, thymus, and peripheral CD4 T cells, and peripheral CD8 T cells from WT mice. The data are representative of three independent experiments. See also *Figure 1—source data 2*.

The online version of this article includes the following source data and figure supplement(s) for figure 1:

**Source data 1.** Sanger sequencing for CYFIP Related Rac1 Interactor A (*Cyria)* and CYFIP Related Rac1 Interactor B (*Cyrib).*

**Source data 2.** Immunoblot for Family with sequence similarity 49 member A (Fam49a) (Cyria) and Family with sequence similarity 49 member B (Fam49b) (Cyrib).

**Figure supplement 1.** Family with sequence similarity 49 member B (Fam49b) expression in thymocyte subsets and T cells from wild-type (WT) mice.

**Figure supplement 1—source data 1.** The numerical data used to generate the *Figure 1—figure supplement 1*.

manner by creating a stop codon in exon 7 of the *Cyria* locus encoding Fam49a (*Figure 1B*). Immunoblot of spleen tissues confirmed that Fam49a or Fam49b expression was undetectable in Fam49a-KO mice or Fam49b-KO mice, respectively in contrast to the wild type (WT, C57BL/6 J) mice (*Figure 1C*). Real-time RT-PCR analysis of flow cytometry-sorted WT thymocytes subsets showed Fam49b is expressed broadly throughout thymic development, whereas Fam49a was mainly expressed in mature T cells (*Figure 1D* and *Figure 1—figure supplement 1*). The expression of Fam49a was not detectable

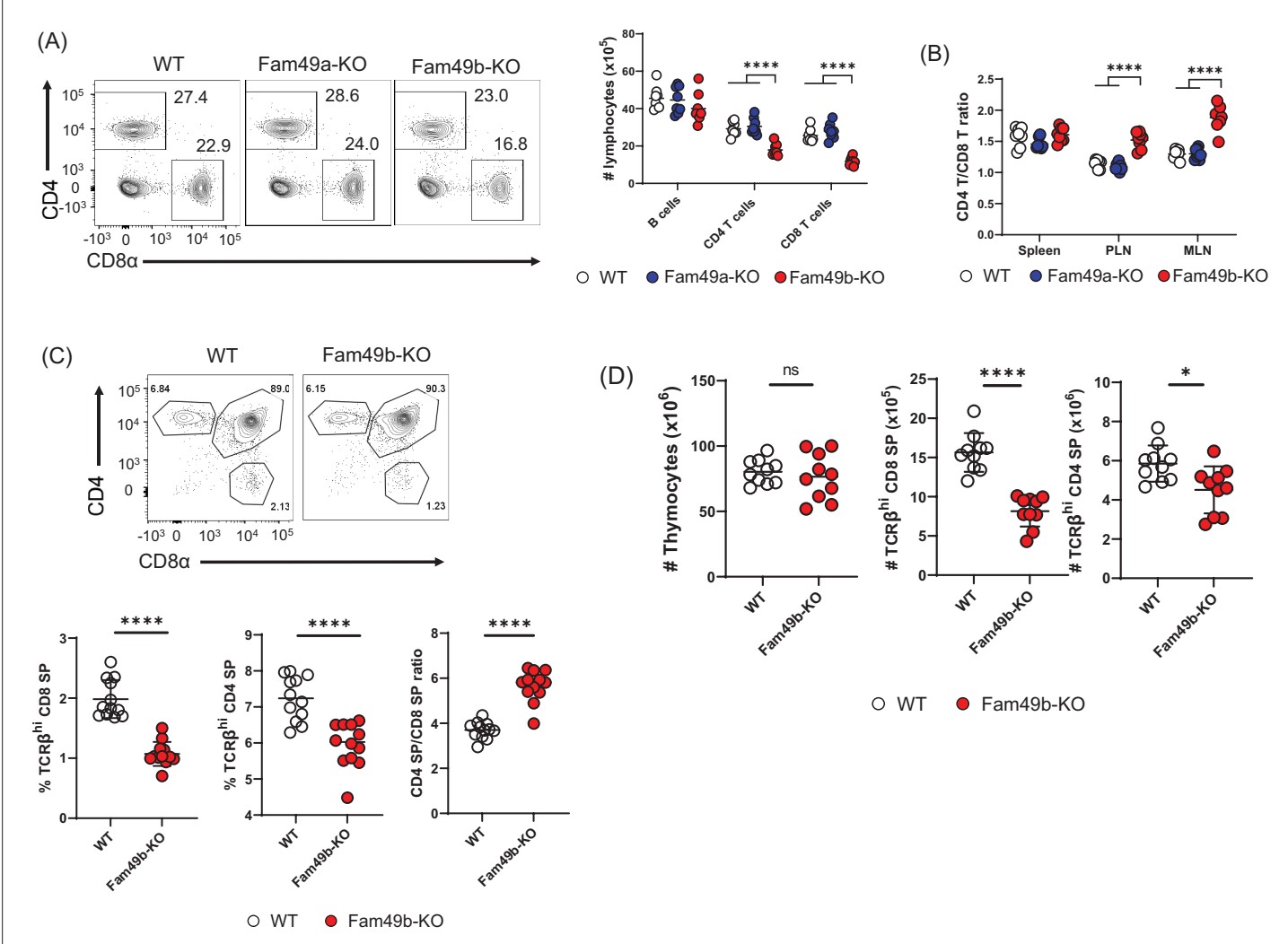

**Figure 2.** Reduced T cell numbers in Family with sequence similarity 49 member B (Fam49b)-KO mice, but not Family with sequence similarity 49 member A (Fam49a)-KO mice. (**A**) Flow cytometry profiles of the expression of CD4 and CD8 (**left**) and the absolute number of lymphocytes in peripheral lymph nodes (**right**) from wild-type WT, Fam49a-KO, and Fam49b-KO mice. Numbers adjust to outlined areas indicate percentage of T cells among total lymphocytes. We used B220 as B cell marker. Each dot represents an individual mouse. Small horizontal lines indicate the mean of 8 mice. ****p<0.0001 (One-way ANOVA). Data are representative of four experiments. See also *Figure 2—source data 1*. (**B**) Ratio of CD4 T cells over CD8 T cells in spleen, peripheral lymph nodes, and mesenteric lymph node in WT, Fam49a-KO, and Fam49b-KO mice. Each dot represents an individual mouse. Small horizontal lines indicate the mean of 8 mice. ****p<0.0001 (One-way ANOVA). Data are representative of four experiments. See also *Figure 2—source data 1*. (**C**) Flow cytometry analyzing the expression of CD4 and CD8 in thymocytes. Contour plots show percentage of CD8 SP and CD4 SP in total thymocytes (**upper**). Percentage of TCRβ+ CD8 SP in total thymocytes (**lower left**), and TCRβ+ CD4 SP in total thymocytes (**lower middle**), and ratio of TCRβ+ CD4 SP cells over TCRβ+ CD8 SP was shown (**lower right**). Each dot represents an individual mouse. Small horizontal lines indicate the mean of 10 mice. ****p<0.0001 (Mann-Whitney test). Data are representative of three experiments. See also *Figure 2—source data 1*. (**D**) Quantification of cell numbers of total thymocytes (**left**), TCRβ+ CD8 SP (**middle**), and TCRβ+ CD4 SP (**right**). Small horizontal lines indicate the mean of 10 mice. *p=0.0124 and ****p<0.0001 (Mann-Whitney test). Data are representative of three experiments. See also *Figure 2—source data 1*.

The online version of this article includes the following source data for figure 2:

**Source data 1.** The numerical data used to generate the *Figure 2*.

in WT thymocytes (*Figure 1D*). Both Fam49a-KO and Fam49b-KO mice were fertile and did not show any apparent abnormalities.

## Defective T cell development in Fam49b-KO mice, but not Fam49a-KO mice

Flow cytometry analysis of cells isolated from lymph nodes showed that the frequency and number of peripheral CD4+ T cells and CD8+ T cells were significantly reduced in Fam49b-KO mice (*Figure 2A*) compared to WT and Fam49a-KO mice. Notably, the reduction in the number of CD8+ T cells was greater than that of CD4+ T cells. As a result, the ratio of CD4+ T cells over CD8+ T cells was increased in Fam49b-KO mice (*Figure 2B*). In contrast, Fam49a-KO mice resembled WT mice in terms of T cell number and CD4/CD8 composition. To further investigate if the decrease in peripheral T cells in Fam49b-KO mice was due to defects of T cell development, we analyzed the surface expression of CD4 and CD8 on thymocytes. The frequencies and numbers of CD4 SP and CD8 SP cells were reduced and the ratios of CD4 SP to CD8 SP thymocytes were increased in Fam49b-KO mice thymi (*Figure 2C* and *Figure 2D*). These data indicate that Fam49b deficiency leads to impaired thymocyte development for both CD4+ and CD8+ T cells, with a more marked impact on the CD8+ T cell population. In contrast, loss of Fam49a showed little, if any, impact on T cell numbers and cellularity in the periphery, or T cell thymic development. Given a lack of any phenotypic changes in Fam49a-KO mice T cells, together with an absence of Fam49a expression in the thymus (*Figure 1D*), we, therefore, focused the remainder of our studies on the Fam49b-KO mice.

## Fam49b-KO thymocytes initiate positive selection but fail to complete development

Successful T cell development is a combined effort of both thymocytes and thymic microenvironment such as thymic epithelial cells and cytokine production. To determine if the effect of Fam49b deficiency on thymocytes development was thymocyte intrinsic or dependent on the extrinsic thymic microenvironment, we generated bone marrow chimeras by injecting WT or Fam49b-KO CD45.2+ bone marrow cells into lethally irradiated WT CD45.1+ mice (B6.SJL-Ptprca Pepcb/BoyJ). A lower frequency and the number of peripheral T cells (*Figure 3A* and *Figure 3—figure supplement 1A–1B*) and increased ratio of peripheral CD4+ T over CD8+ T was observed in Fam49b-KO chimera mice compared to WT chimera mice (*Figure 3B*). The Fam49b-KO thymocytes developed in WT thymic environment are like those developed in the germline Fam49b-KO environment in terms of both thymocyte and peripheral lymphocyte phenotypes (*Figure 3—figure supplement 2A–2C*). Therefore, the effect of Fam49b mutation on T cell development is predominantly due to thymocyte intrinsic functions.

Next, we sought to determine which step of T cell development was altered in Fam49b-KO mice. We thus subdivided thymocytes into four stages based on the differential expression of TCRβ and CD69 expression (*Figure 3C* and *Figure 3—figure supplement 3*; *Hu et al., 2012*). The proportion of stage 1 thymocytes (TCRβ^lo^CD69^-^), which include the DN and pre-selection DP cells, were similar between WT and Fam49b-KO mice. The percentage of stage 2 thymocytes (TCRβ^int^CD69^+^), which represent transitional DP undergoing TCR-mediated positive selection, were significantly higher in the Fam49b-KO mice. The proportion of late-stage thymocytes including the post-positive selection (TCRβ^hi^CD69^+^) and the mature thymocytes (TCRβ^hi^CD69^-^) were markedly decreased (*Figure 3C*). Consistent with our observation in the periphery, increased ratios of CD4 SP to CD8 SP were observed among the late-stage thymocytes (TCRβ^hi^CD69^+^ and TCRβ^hi^CD69^-^) in Fam49b-KO mice (*Figure 3D*). These data show that the post-positive-selection process is impaired in Fam49b-KO thymocytes.

We further distinguished the pre-and post-positive selection populations by expression of cell surface TCRβ and CD5 (*Figure 3E* and *Figure 3—figure supplement 4*; *Hu et al., 2012*). These markers define a developmental progression: stage 1 (TCRβ^lo^CD5^lo^) represents the pre-selection phase of DP thymocytes, and Stage 2 (TCRβ^lo^CD5^int^) are cells initiating positive selection. Stage 3 (TCRβ^int^CD5^hi^) represents thymocytes in the process of undergoing positive selection, and Stage 4 (TCRβ^hi^CD5^hi^) consists primarily of post-positive selection SP thymocytes. We observed that all the early phase populations (TCRβ^lo^CD5^lo^, TCRβ^lo^CD5^int^, TCRβ^int^CD5^hi^) increased significantly in proportion in Fam49b-KO, whereas the post-positive selection SP thymocytes (TCRβ^hi^CD5^hi^) were markedly decreased (*Figure 3E*). Similarly, an increased ratio of CD4 SP to CD8 SP was observed in the post-positive selection population (TCRβ^hi^CD5^hi^) in Fam49b-KO thymocytes (*Figure 3F*). This phenotype was

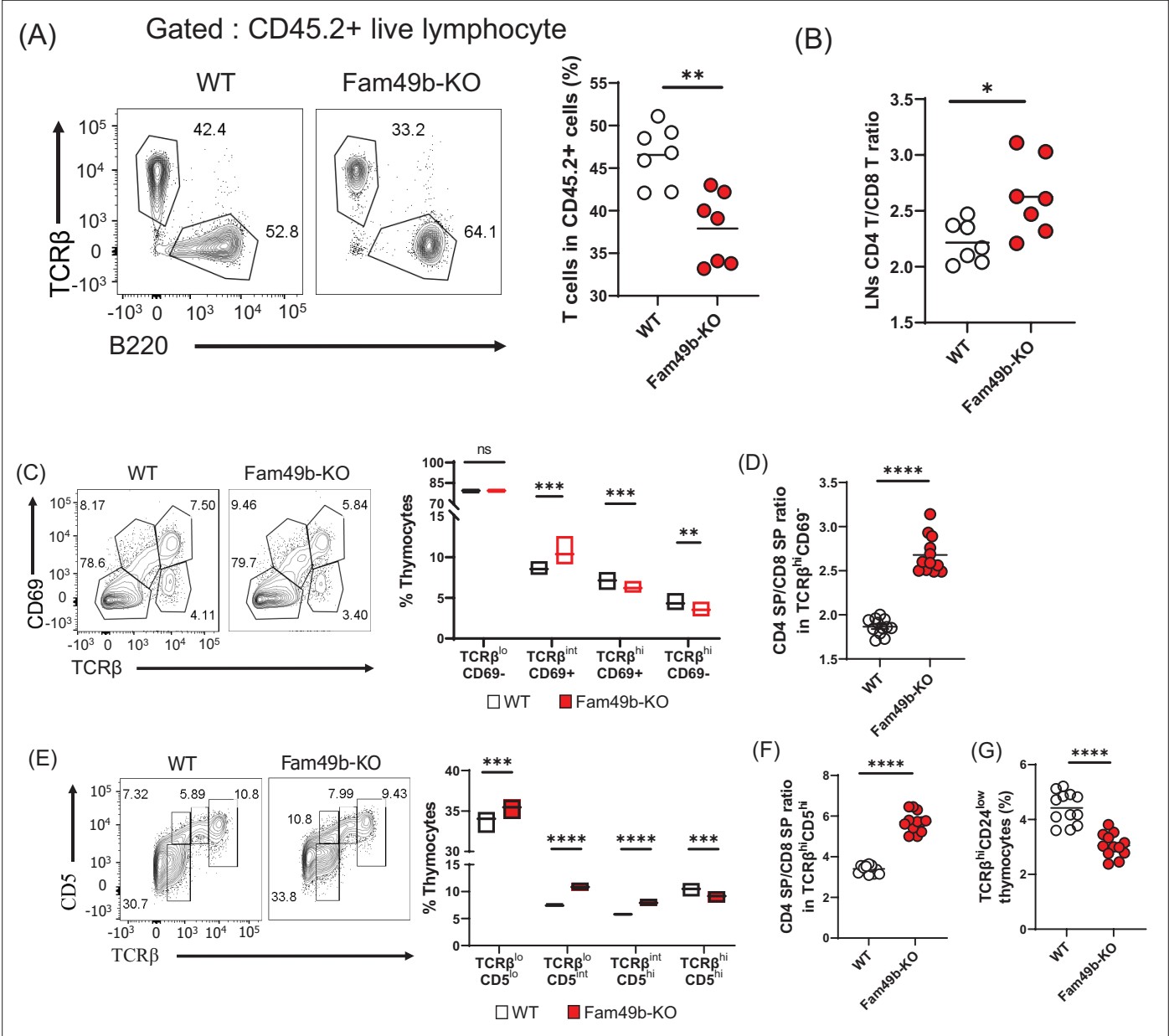

**Figure 3.** Defective thymic development in Family with sequence similarity 49 member B (Fam49b)-KO mice. (**A**) Expression of TCRβ and B220 expressing cells (**left**) and frequency of TCRβ expressing cells among CD45.2+ total lymph node cells from bone marrow chimera mice (**right**). Bone marrow from either wild-type (WT) or Fam49b-KO mice was injected i.v. into lethally irradiated CD45.1+ WT mice and chimeric mice were analyzed 8 weeks later. Small horizontal lines indicate the mean of 7 mice. **=0.0047 (Mann-Whitney test). Data are pooled from two independent experiments. See also *Figure 3—source data 1*. (**B**) Ratio of CD4 T cells over CD8 T cells in CD45.2+ total lymph node cells from bone marrow chimera mice. Bone marrow from either WT or Fam49b-KO mice was injected i.v. into lethally irradiated CD45.1+ WT mice and chimeric mice were analyzed 8 weeks later. Small horizontal lines indicate the mean of 7 mice. *p=0.0192 (Mann-Whitney test). Data are pooled from two independent experiments. See also *Figure 3—source data 1*. (**C**) (**left**) Differential surface expression of CD69 and TCRβ was used to identify thymocyte population of different maturity in WT and Fam49b-KO mice. (**right**) Dot Plots show percentages of different thymocyte subpopulations in WT and Fam49b-KO mice. Numbers adjust to outlined areas indicate percentage of thymocytes subset among total thymocytes. Floating bars (min to max). horizontal lines indicate the mean of 12 mice. **p=0.0038 and ***p=0.0003 and ***p=0.0001 (Mann-Whitney test). Data are representative of five experiments. See also *Figure 3—source data 1*. (**D**) Ratio of CD4 single positive (SP) cells over CD8 SP cells in TCRβhiCD69- thymocyte subpopulation. horizontal lines indicate the mean of 12 mice. ****p<0.0001 (Mann-Whitney test). Data are representative of five experiments. See also *Figure 3—source data 1*. (**E**) (**left**) Differential surface expression of CD5 and TCRβ was used to identify thymocyte population of different maturity in WT and Fam49b-KO mice. (**right**) Dot Plots show percentages of different thymocyte subpopulations from mice. Numbers adjust to outlined areas indicate percentage of thymocytes subset among total thymocytes. Floating bars (min to max). Horizontal lines indicate the mean of 12 mice. ***p=0.0005 and ***p=0.0002 and ****p<0.0001

*Figure 3 continued on next page*

*Figure 3 continued*

(Mann-Whitney test). Data are representative of five experiments. See also *Figure 3—source data 1*. (**F**) Ratio of CD4 SP cells over CD8 SP cells in TCRβ$^{hi}$CD5$^{hi}$ thymocyte subpopulation. Small horizontal lines indicate the mean of 12 mice. ****p<0.0001 (Mann-Whitney test). Data are representative of five experiments. See also *Figure 3—source data 1*. (**G**) Frequency of TCRβ$^{hi}$CD24$^{low}$ thymocyte subpopulation among total live thymocytes. Small horizontal lines indicate the mean of 12 mice. ****p<0.0001 (Mann-Whitney test). Data are representative of five experiments. See also *Figure 3—source data 1*.

The online version of this article includes the following source data and figure supplement(s) for figure 3:

**Source data 1.** The numerical data used to generate the *Figure 3*.

**Figure supplement 1.** Lower frequencies and numbers of T cells in Family with sequence similarity 49 member B (Fam49b)-KO chimera mice.

**Figure supplement 1—source data 1.** The numerical data used to generate the *Figure 3—figure supplement 1*.

**Figure supplement 2.** Analyzing thymocytes in Family with sequence similarity 49 member B (Fam49b)-KO chimera mice.

**Figure supplement 2—source data 1.** The numerical data used to generate the *Figure 3—figure supplement 2*.

**Figure supplement 3.** Analyzing thymic selection using TCRβ and CD69 expression in thymus.

**Figure supplement 4.** Analyzing thymic selection using TCRβ and CD5 expression in thymus.

further verified by the observation of lower percentage of mature SP CD24$^{lo}$TCRβ$^{hi}$ cells in Fam49b-KO mice compared with WT mice (*Figure 3G*). Taken together, these results suggest that Fam49b plays an important role in T cells development, especially in TCRβ$^{hi}$CD69$^+$ and TCRβ$^{hi}$CD69$^-$ thymocytes.

## Enhanced negative selection in Fam49b-KO thymocytes

Based on our observation that the loss of Fam49b led to decreased mature thymocyte populations, together with evidence that Fam49b can negatively regulate TCR signaling (*Shang et al., 2018*), we hypothesized that enhanced clonal deletion due to elevated TCR signaling strength would lead to the loss of positively selected thymocytes in Fam49b-KO mice. To test this hypothesis, we assessed the cleavage of caspase 3, one of the key apoptosis events during clonal deletion (*Figure 4—figure supplement 1*; *Breed et al., 2019*). In the thymus, caspase 3 is cleaved in the apoptotic cells due to either clonal deletion (i.e. negative selection) or death by neglect (i.e. failed positive selection). To distinguish between these two fates, we stained the cells for TCRβ and CD5 molecules which are upregulated upon TCR stimulation. Thus, cleaved-caspase3$^+$TCRβ$^{hi}$CD5$^{hi}$ cells represent thymocytes undergoing clonal deletion, whereas cleaved-caspase 3$^+$TCRβ$^-$CD5$^-$ cells represent thymocytes undergoing death by neglect. We observed that the frequency of cells undergoing clonal deletion was increased among Fam49b-KO thymocytes, whereas the frequencies of cells to be eliminated through death by neglect were similar between Fam49b-KO and WT mice (*Figure 4A*).

Negative selection can occur in the thymic cortex as DP thymocytes are undergoing positive selection or in the thymic medulla after positive selection (*McCaughtry et al., 2008*). To determine whether loss of Fam49b led to increased deletion in the cortex or medulla, we stained the thymocytes for CCR7, which marks medullary thymocytes and is the receptor for the medullary chemokines CCL19/21 (*Breed et al., 2019*; *Ueno et al., 2004*). The frequencies of cleaved-caspase3$^+$CCR7$^-$ cells and cleaved-caspase3$^+$CCR7$^+$ cells were significantly increased in the Fam49b-KO mice, suggesting that more thymocytes were eliminated through clonal deletion in both the cortex and medulla of Fam49b-KO thymus as compared with WT thymus (*Figure 4A*).

Next, to determine if TCR-signal strength in Fam49b-KO thymocyte was increased, we assessed the surface expression of CD5, a surrogate marker for TCR-signal strength (*Tarakhovsky et al., 1995*; *Azzam et al., 1998*). We found that CD5 expression was upregulated on Fam49b-KO DP thymocytes, but not on CD4 SP and CD8 SP thymocytes (*Figure 4B*), suggesting Fam49b-KO DP thymocytes had received stronger TCR signaling than the WT thymocytes. Fam49b has been identified as an inhibitor of TCR signaling via the Rac-PAK axis in Fam49b-KO Jurkat T cells (*Shang et al., 2018*). Thus, we assess the activation of key TCR signaling cascade components in total thymocytes of WT and Fam49b-KO mice after TCR stimulation with anti-CD3ε and CD4 mAb. Fam49b deficiency led to prolonged increases in all of the downstream phosphorylation events tested, including ZAP-70, LAT, PLCγ1, and ERK (*Figure 4C*). PAK phosphorylation was also dramatically elevated in Fam49b-KO thymocytes (*Figure 4D*). In summary, enhanced TCR-signaling strength intrinsic to Fam49b-KO DP thymocytes led to excessive clonal deletion in the cortex and medulla, resulting in the loss of naïve mature T cells in both thymus and periphery in the mice.

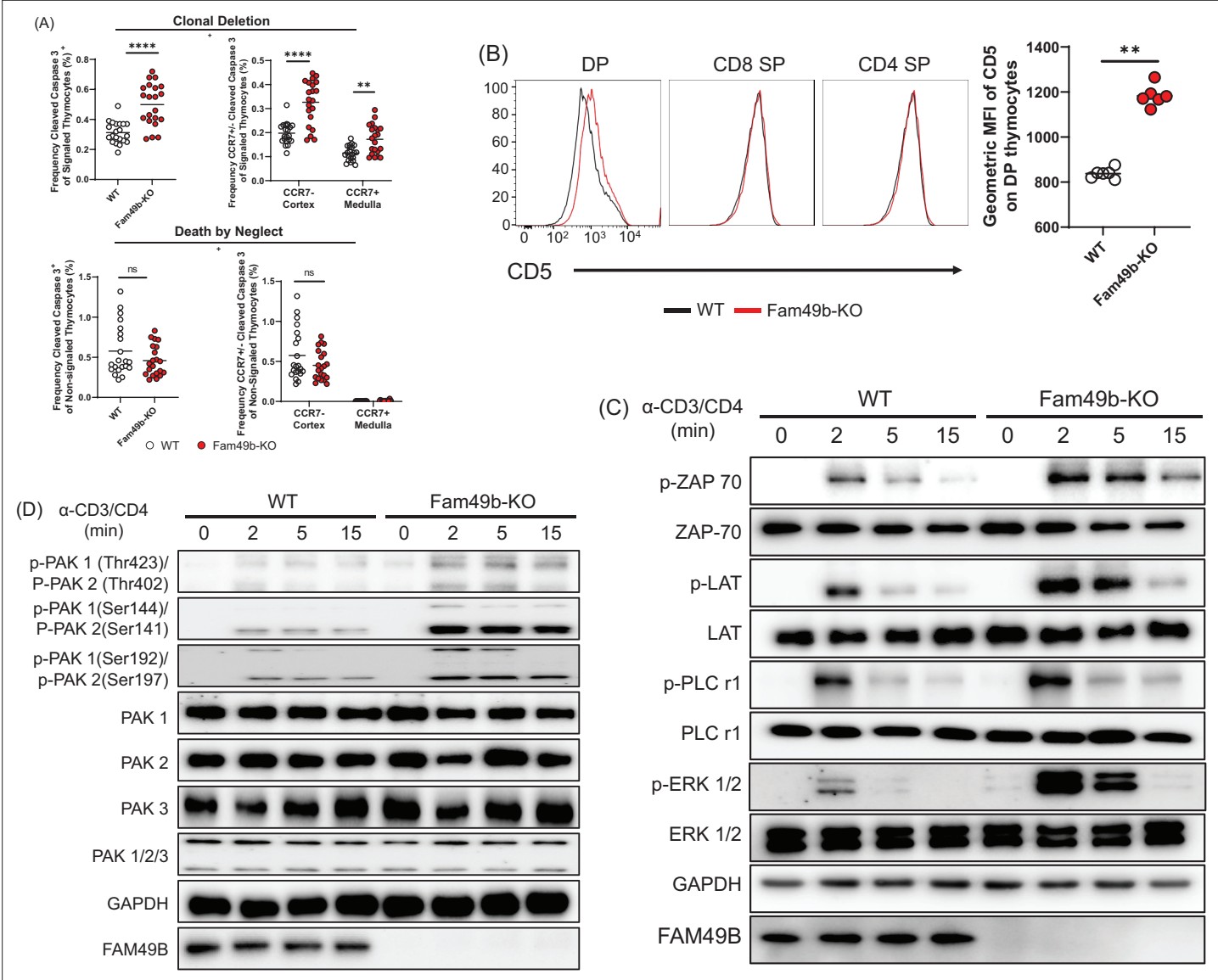

**Figure 4.** Enhanced negative selection due to elevated T cell receptor (TCR) signaling in Family with sequence similarity 49 member B (Fam49b)-KO thymocytes. (**A**) Frequency of cleaved caspase 3+ cells among TCRβhiCD5hi (Signaled, **upper left**) and TCRβ-CD5- (Non-signaled, **lower left**) thymocytes. Frequency of CCR7+ cleaved caspase 3+ and CCR7- cleaved caspase 3+ cells among TCRβhiCD5hi (Signaled, **upper right**) and TCRβ-CD5- (Non-signaled, **lower right**) thymocytes. Small horizontal lines indicate the mean of 21 mice. **p=0.0017 and ****p<0.0001 (Mann-Whitney test). Data are pooled from three independent experiments. See also *Figure 4—source data 1*. (**B**) Expression of activation marker CD5 on double positive (DP), TCRβ+ CD4 single positive (SP), and TCRβ+ CD8 SP thymocytes from WT and Fam49b-KO mice (**left**). Geometric MFI of CD5 on DP thymocytes (**right**). Small horizontal lines indicate the mean of 6 mice. **p=0.0022 (Mann-Whitney test). Data are representative of seven experiments. See also *Figure 4—source data 1*. (**C**) Immunoblot analysis of TCR cascade component activation in total thymocytes from WT or Fan49b-KO mice. Total thymocytes were stimulated with soluble anti-CD3ε and anti-CD4 antibodies for the times indicated. The data are representative of three independent experiments. See also *Figure 4—source data 2*. (**D**) Immunoblot analysis of PAK cascade component activation in total thymocytes from WT or Fan49b-KO mice. The data are representative of four independent experiments. See also *Figure 4—source data 3*.

The online version of this article includes the following source data and figure supplement(s) for figure 4:

**Source data 1.** The numerical data used to generate the *Figure 4*.

**Source data 2.** Immunoblot for T cell receptor (TCR) signaling.

**Source data 3.** Immunoblot for PAK signaling.

**Figure supplement 1.** Flow cytometry gating strategies to measure clonal deletion and death by neglect.

## Decreased survival rate of Fam49b-KO SP thymocytes

To investigate the effect of Fam49b deficiency at later stages of thymocytes development, we analyzed the surface expression of CD69 and CD62L on SP thymocytes. SP thymocytes in the thymus can be divided into immature (CD69$^{hi}$CD62L$^{lo}$), semi-mature (CD69$^{lo}$CD62L$^{lo}$), and mature (CD69$^{lo}$CD62L$^{hi}$) SP subpopulation (*Phee et al., 2014*). Consistent with excessive clonal deletion in the cortex and medulla (*Figure 4A*), the absolute numbers of immature and mature CD8 SP and CD4 SP were decreased from Fam49b-KO mice (*Figure 5A* and *Figure 5B*). Expression of IL-7Rα was increased in post-positive selection DP and SP thymocytes, while most DP thymocytes down-regulate expression of IL-7Rα (*Van De Wiele et al., 2004*). We observed that IL-7Rα expression was significantly decreased in some Fam49b-KO CD4 SP thymocytes, suggesting Fam49b deficiency could affect survival of SP thymocytes (*Figure 5C*). To determine whether loss of Fam49b increased cell death or apoptosis in SP thymocytes, we stained the SP thymocytes for 7-AAD and Annexin V to detect apoptotic and necrotic cells. A twofold increase in cell death was observed in Fam49b-deficient immature and semi-mature SP thymocytes, whereas the frequency of dead cells detected among mature SP thymocytes was unaffected (*Figure 5D* and *Figure 5E* and *Figure 5—figure supplement 1A–1B*). Our results suggest that Fam49b plays a key role in the maturation and maintenance of immature and semi-mature SP thymocytes, but not in the survival of mature SP thymocytes.

## Lower peripheral T cell survival of Fam49b-KO OT-I CD8 T cells

To explore thymocyte development in more detail, we crossed the Fam49b-KO mice onto the OT-I TCR-transgenic strain. Based on our observation that enhanced TCR-signaling strength intrinsic to Fam49b-KO DP thymocytes leads to excessive clonal deletion in the cortex and medulla (*Figure 4A* and *Figure 4B*), we postulated that enhanced TCR-signal strength of Fam49b-KO OT-I thymocytes could be diverted into negative selection from positive selection. However, contrary to the predictions, there were no significant differences in frequencies and numbers of CD8 SP thymocytes in Fam49b-KO OT-I mice at 6 weeks (*Figure 6A* and *Figure 6B*). Numbers of total CD8 T cells and naïve CD8 T cells were, again, normal in 6 weeks old Fam49b-KO OT-I mice. Interestingly, however, the numbers of total CD8 T cells and naïve CD8 T cells were significantly decreased in Fam49b-KO OT-I mice at 12 weeks. (*Figure 6C* and *Figure 6D* and *Figure 6—figure supplement 1A*). Severe T cell deficiency in the periphery of Fam49b-KO OT-I mice at 12 weeks was at odds with the similar numbers of peripheral CD8 T cells of Fam49b-KO OT-I mice at 6 weeks. We hypothesized that Fam49b may play a key role in peripheral T cell survival. To determine whether loss of Fam49b increased cell death in naïve CD8 T cells, we stained the naïve CD8 T cells for 7-AAD and Annexin V. We found that the frequency of Annexin V$^+$ 7-AAD$^+$ naïve CD8 T cells were significantly increased in spleen and lymph nodes from Fam49b-KO OT-I mice as well as Fam49b-KO mice (*Figure 6E* and *Figure 6—figure supplement 1B*), suggesting Fam49b deficiency promotes naïve CD8 T cells death in the peripheral. BIM/Bcl-2 balance is essential for controlling the homeostasis of naïve and memory T cells (*Wojciechowski et al., 2007*). Therefore, we measured the expression of BIM and Bcl-2 molecules in naïve T cells and total thymocytes from WT and Fam49b-KO mice. We observed that a higher ratio of BIM/Bcl-2 is detected in naïve T cells as well as total thymocytes (*Figure 6—figure supplement 2A–B*). In summary, these data show that Fam49b is required for peripheral T cells survival and maintenance.

## Impaired development of natural IELs in Fam49b-KO mice

Some self-reactive thymocytes rely on strong TCR signaling to mature into unconventional T cell subsets through utilizing an alternative selection process known as agonist selection (*Oh-Hora et al., 2013*; *Hogquist and Jameson, 2014*). Due to the robust effects of Fam49b deficiency on TCR-signaling strength, we investigated whether Fam49b affects the development of well-known agonist-selected T cell subsets including CD8αα$^+$TCRαβ$^+$ IELs in small intestinal epithelium, iNKT cells in liver, and Treg cells in lymph nodes (*Lambolez et al., 2007*; *Kronenberg and Gapin, 2002*; *Hsieh et al., 2012*). We found that all three T cell subsets were differentially affected by the loss of Fam49b. The percentage of CD8αα$^+$TCRαβ$^+$ IELs among IEL T cells was significantly decreased from 60% in WT mice to 30% in Fam49b-KO mice, whereas the frequency of liver iNKT cells was unaffected (*Figure 7A* and *Figure 7—figure supplement 1*). The frequency of Treg among lymph node CD4$^+$ T cells increased slightly from 16 to 20% in lymph nodes in Fam49b-KO mice, though the absolute number of Treg

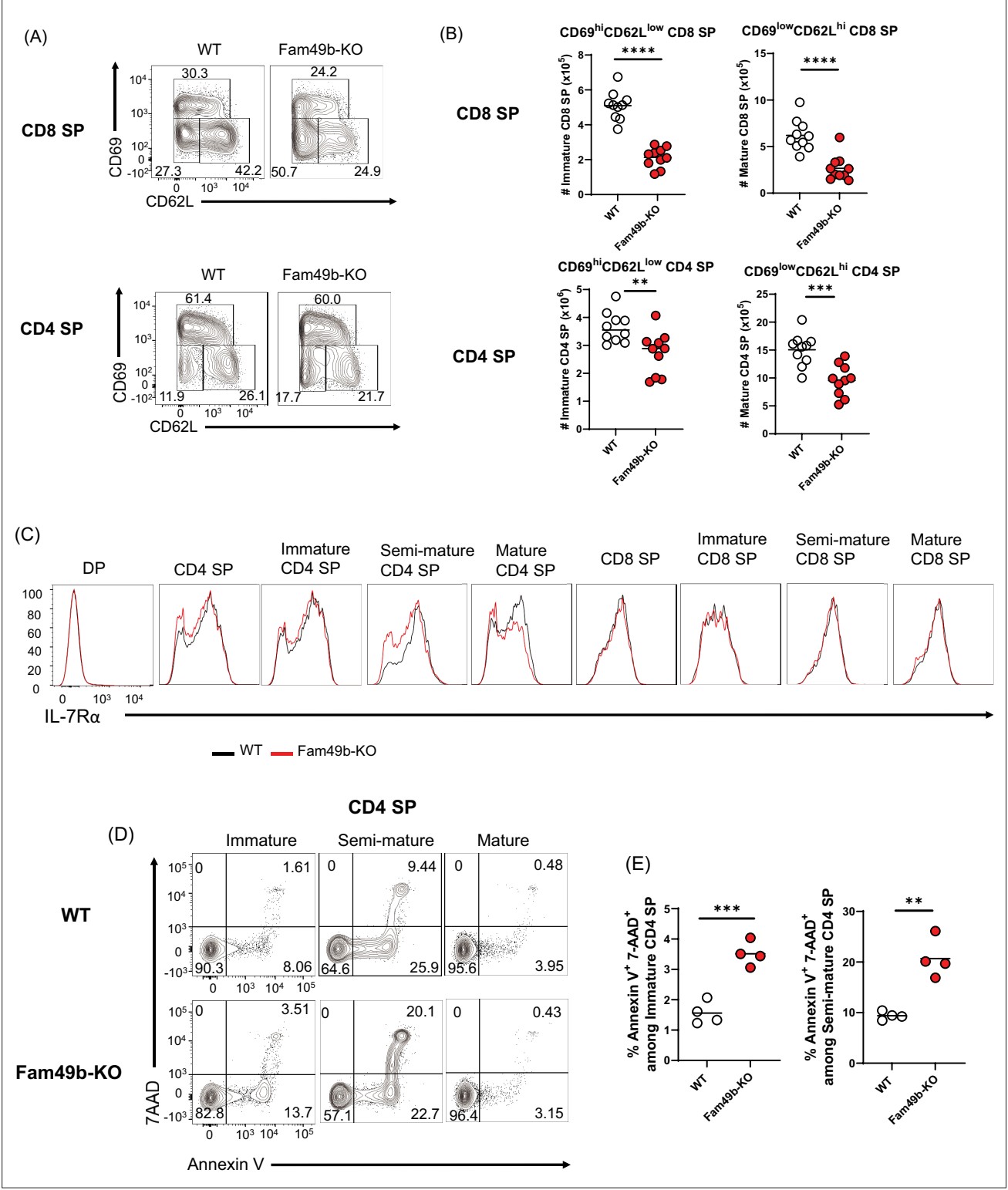

**Figure 5.** Elevated apoptosis in Family with sequence similarity 49 member B (Fam49b)-KO single positive (SP) thymocytes. (**A**) Frequencies of immature (CD62L$^{lo}$CD69$^{hi}$), semi-mature (CD62$^{lo}$CD69$^{lo}$), and mature (CD62$^{hi}$CD69$^{lo}$) in the TCRβ$^{hi}$ CD8 SP (**upper**) and TCRβ$^{hi}$ CD4 SP thymocytes (**lower**) from wild-type (WT) and Fam49b-KO mice. Numbers adjust to outlined areas indicate percentage of each population among total SP TCRβ$^+$ thymocytes. Shown are representative data of ten mice per genotype. (**B**) Quantification of cell numbers of immature and mature fraction in TCRβ$^{hi}$ CD8 SP thymocytes (**upper**) and TCRβ$^{hi}$ CD4 SP thymocytes (**lower**). Each dot represents an individual mouse. Horizontal lines indicate the mean of 10 mice. \*\*p=0.0069 and \*\*\*p=0.0004 and \*\*\*\*p<0.0001 (Mann-Whitney test). Data are representative of three experiments. See also *Figure 5—source data*

*Figure 5 continued on next page*

*Figure 5 continued*

**1** (**C**) Decreased expression of IL-7Rα on double positive (DP), total CD4 SP, immature CD4 SP, semi-mature CD4 SP, mature CD4 SP, total CD8 SP, immature CD8 SP, semi-mature CD8 SP, and mature CD8 SP thymocytes from WT and Fam49b-KO mice. Shown are representative data of ten mice per genotype. (**D**) Apoptosis detected using 7AAD/Annexin V staining in freshly isolated Immature (CD62L$^{lo}$CD69$^{hi}$), semi-mature (CD62$^{lo}$CD69$^{lo}$), and mature (CD62$^{hi}$CD69$^{lo}$) TCRβ$^{hi}$ CD4 SP thymocytes from WT (**upper**) and Fam49b-KO mice (**lower**). Shown are representative data of two independent experiments. (**E**) Frequency of 7AAD$^+$ and Annexin V$^+$ cells among immature CD4 SP (**left**) and semi-mature CD4 SP (**right**) thymocytes from WT and Fam49b-KO mice. Small horizontal lines indicate the mean of 4 mice. **p=0.0084 and ***p=0.0004 (Mann-Whitney test). Data are pooled from two independent experiments. See also *Figure 5—source data 1*.

The online version of this article includes the following source data and figure supplement(s) for figure 5:

**Source data 1.** The numerical data used to generate the *Figure 5*.

**Figure supplement 1.** Elevated apoptosis in Family with sequence similarity 49 member B (Fam49b)-KO CD8 single positive (SP) thymocytes and peripheral T cells.

**Figure supplement 1—source data 1.** The numerical data used to generate the *Figure 5—figure supplement 1*.

---

was ~80% of the number in WT mice. Enhanced frequency of Treg seems to be a result of a greater reduction of total CD4$^+$ T cells compared to Treg (*Figure 7A* and *Figure 7—figure supplement 2A*).

Gut IEL T lymphocytes are extremely heterogenous, and based on the differentiation mechanisms, can be subdivided into two major subpopulations including natural intraepithelial lymphocytes (natural IELs) and induced intraepithelial lymphocytes (induced IELs) (*Cheroutre et al., 2011*). Natural IELs are home to the gut immediately after thymic maturation. They are TCRγδ$^+$ and TCRαβ$^+$ T cells that can be either CD8αα$^+$ or CD8αα$^-$. In contrast, induced IELs arise from conventional peripheral CD8αβ$^+$TCRαβ$^+$ T cells and are activated post-thymically in response to peripheral antigens. The two populations can be distinguished by the expression of CD5; natural IELs are CD5$^-$ and induced IELs CD5$^+$ (*Figure 7—figure supplement 1*). Based on our observation of the dramatic loss of CD8αα$^+$TCRαβ$^+$ IELs in Fam49b-KO mice, we postulated that other IEL subsets might be altered as well. Fam49b-KO mice showed a substantial reduction of natural IELs, including both the TCRγδ$^+$ IELs as well as CD8αα$^+$TCRαβ$^+$ IELs (*Figure 7B*), whereas the relative frequencies of induced IELs (CD8αβ$^+$TCRαβ$^+$ IELs) were increased (*Figure 7C* and *Figure 7—figure supplement 2B*). These results suggest that Fam49b is involved in shaping the agonist-selected unconventional T cell populations and that Fam49b deficiency leads to substantial loss of the natural IELs, including CD8αα$^+$TCRαβ$^+$ IELs and TCRγδ$^+$ IELs.

CD8αα$^+$TCRαβ$^+$ IELs develop from two distinct CD8αα IEL precursors (IELps) in the thymus (*Ruscher et al., 2017*; *Ruscher et al., 2020*). CD8αα IELps include PD-1$^+$ type A and PD-1$^-$ type B IELps populations. Type A IELps localized to the cortex and were integrin α4β7$^+$, while type B IELps localized to the medulla and expressed CD103. Based on our observation of the dramatic loss of CD8αα$^+$TCRαβ$^+$ IELs in Fam49b-KO mice, we postulated that two CD8αα IELps might be altered in the thymus. Contrary to our expectations, there was no decrease in the frequencies and numbers of two CD8αα IELps in the thymus (*Figure 7—figure supplement 3A*). The integrin expression of each CD8αα IELps was also comparable between WT and Fam49b-KO mice (*Figure 7—figure supplement 3B–3C*). Severe CD8αα$^+$TCRαβ$^+$ IELs deficiency in the gut was at odds with the higher number of thymic Type B IELps. We found that Fam49b is also required for the maintenance and survival of peripheral T cells (*Figure 6* and *Figure 6—figure supplement 1*). Therefore, we reasoned that the lower number of CD8αα$^+$TCRαβ$^+$ IELs may be the result of impaired maintenance or survival in the periphery, rather than a problem with thymic IELps development.

## Discussion

Development of T cells is critically dependent on the strength of signaling through the TCR that leads to positive or negative selection (*Gaud et al., 2018*; *Hwang et al., 2020*). However, the roles of additional intracellular proteins and signaling pathways that regulate TCR signaling strength in the thymus have not been fully elucidated. Here, by studying the thymic development of T cells in Fam49b-KO mice, we report that Fam49b finetunes thymic selection by negatively regulating TCR signal-strength in the thymus and is essential for normal thymocyte development. Mice deficient in Fam49b developed severe T cell lymphopenia due to enhanced TCR-signaling in DP thymocytes. In Fam49b-KO thymus, the post-positively selected population was significantly reduced, while the generation of

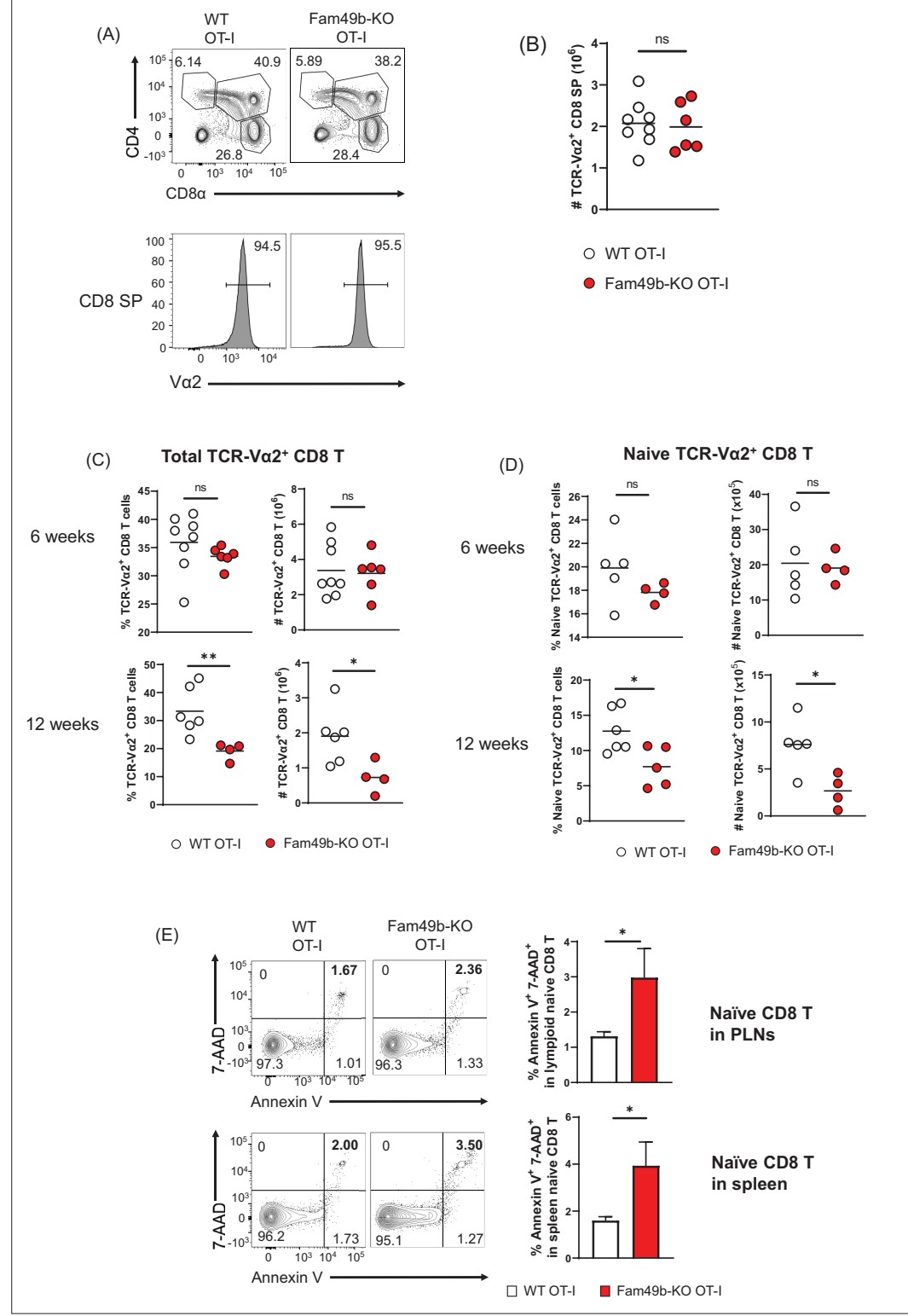

**Figure 6.** Dramatic loss of peripheral T cells, but not thymocytes, in Family with sequence similarity 49 member B (Fam49b)-KO OT-I mice with age. (**A**) Flow cytometry analyzing the expression of CD4 and CD8 in thymocytes from OT-I and Fam49b-KO OT-I mice at 6 weeks of age (**upper**). Contour plots show percentage of DP, CD8 single positive (SP), and CD4 SP among total thymocytes. Staining with antibody to the OT-I-specific variable region Vα2 (**lower**). Numbers above outlined areas indicate percentage of TCR-Vα2+ cells among CD8 SP thymocytes. Shown are representative data of 6–8 mice per genotype. (**B**) Quantification of cell numbers of TCR-Vα2+ CD8 SP

*Figure 6 continued on next page*

*Figure 6 continued*

thymocytes from OT-I and Fam49b-KO OT-I mice at 6 weeks of age. Small horizontal lines indicate the mean of 6–8 mice. Data are pooled from two independent experiments. See also *Figure 6—source data 1*. (**C**) Percentage of TCR-Vα2$^+$ CD8 T cells in lymph nodes from OT-I and Fam49b-KO OT-I mice at either 6 weeks (**upper left**) or 12 weeks (**lower left**) of age. Quantification of cell numbers of TCR-Vα2$^+$ CD8 T cells in peripheral lymph nodes from OT-I and Fam49b-KO OT-I mice at either 6 weeks (**upper right**) or 12 weeks (**lower right**) of age. Small horizontal lines indicate the mean of 4–8 mice. Each dot represents an individual mouse. *p=0.0173 and **p=0.0077 (Mann-Whitney test). Data are pooled from three independent experiments. See also *Figure 6— source data 1*. (**D**) Percentage of TCR-Vα2$^+$ CD62L$^+$CD44$^{lo}$ CD8 T cells in peripheral lymph nodes from OT-I and Fam49b-KO OT-I mice at either 6 weeks (**upper left**) or 12 weeks (**lower left**) of age. Quantification of cell numbers of TCR-Vα2$^+$ CD62L$^+$CD44$^{lo}$ CD8 T cells in peripheral lymph nodes from OT-I and Fam49b-KO OT-I mice at either 6 weeks (**upper right**) or 12 weeks (**lower right**) of age. Small horizontal lines indicate the mean of 4–5 mice. Each dot represents an individual mouse. *p=0.0154 or 0.0204 (Mann-Whitney test). Data are pooled from two independent experiments. See also *Figure 6—source data 1*. (**E**) Apoptosis detected using 7AAD/Annexin V staining in fleshly isolated CD62L$^+$CD44$^{lo}$ CD8 T cells in peripheral lymph nodes (top) and spleen (**lower**) from OT-I and Fam49b-KO OT-I mice at 16 weeks of age. Shown are representative data of 4 mice per genotype. Right panels show average frequencies of 7AAD$^+$ and Annexin V$^+$ cells among CD62L$^+$CD44$^{lo}$ CD8 T cells in peripheral lymph nodes (mean and SEM, n=4) and spleen (mean and SEM, n=4). *p=0.0286 (Mann-Whitney test). Data are pooled from two independent experiments. See also *Figure 6—source data 1*.

The online version of this article includes the following source data and figure supplement(s) for figure 6:

**Source data 1.** The numerical data used to generate the *Figure 6*.

**Figure supplement 1.** Dramatic loss of CD44$^{lo}$ peripheral T cells in Family with sequence similarity 49 member B (Fam49b)-KO OT-I mice at 12 weeks.

**Figure supplement 1—source data 1.** The numerical data used to generate the *Figure 6—figure supplement 1*.

**Figure supplement 2.** Elevated ratio of BIM to Bcl-2 in Family with sequence similarity 49 member B (Fam49b)-KO mice compared to wild-type (WT) mice.

**Figure supplement 2—source data 1.** Immunoblot for BIM/Bcl-2.

**Figure supplement 2—source data 2.** The numerical data used to generate the *Figure 6—figure supplement 2*.

---

DN or immature DP thymocytes was mostly unaffected. We further confirmed that the loss of post-positive selection thymocytes in Fam49b-KO mice was due to enhanced clonal deletion instead of death by neglect. As a result, the frequencies of CD4 SP and CD8 SP cells in the Fam49b-KO thymi were significantly reduced.

While the medulla is a specialized site for negative selection, a substantial amount of negative selection occurs in the thymic cortex, overlapping in space and time with positive selection (*Stritesky et al., 2013*; *McDonald et al., 2015*). We found that the frequency of the thymocytes undergoing clonal deletion was significantly increased in Fam49b-KO thymus, while the frequency of thymocytes undergoing death by neglect remained the same. Moreover, most of thymocytes undergoing clonal deletion were CCR7$^-$ cortex resident thymocytes (~65%) in both WT and Fam49b-KO thymus. These data imply that Fam49b is needed immediately after the initial positive selection stage to serve as a 'brake' which dampens TCR signaling, thus helping to avoid negative selection. This 'brake,' once taken out of the picture, leads to overexuberant clonal deletion and subsequent loss of a large proportion of the mature T cells.

Fam49b-KO DP thymocytes received a stronger TCR signal compared to WT DP thymocytes. At the molecular level, Fam49b directly interacts with active Rac and negatively regulates its activity (*Shang et al., 2018*; *Fort et al., 2018*; *Yuki et al., 2019*). Rac plays key roles in cytoskeleton remodeling, signal transduction, and regulation of gene expression in thymocytes and peripheral T cells (*Saoudi et al., 2014*; *Bosco et al., 2009*). The modulation of Rac activity by switching between its two conformational states, i.e., inactive (GDP-bound) and active (GTP-bound), is essential for multiple stages of thymocyte development and maturation. Previous studies suggested that Rac activity is important for β selection at DN thymocytes as well as positive and negative selection at DP thymocytes (*Dumont et al., 2009*; *Saoudi et al., 2014*). Moreover, transgenic mice that express constitutively active Rac-1 mutant revealed that Rac-1 activity could reverse the fate of thymocytes from positive to negative selection in the thymus (*Gomez et al., 2001*). Taken together, the phenotype similarities between active Rac-1 transgenic and our Fam49b-KO mice, and the association between Rac and Fam49b

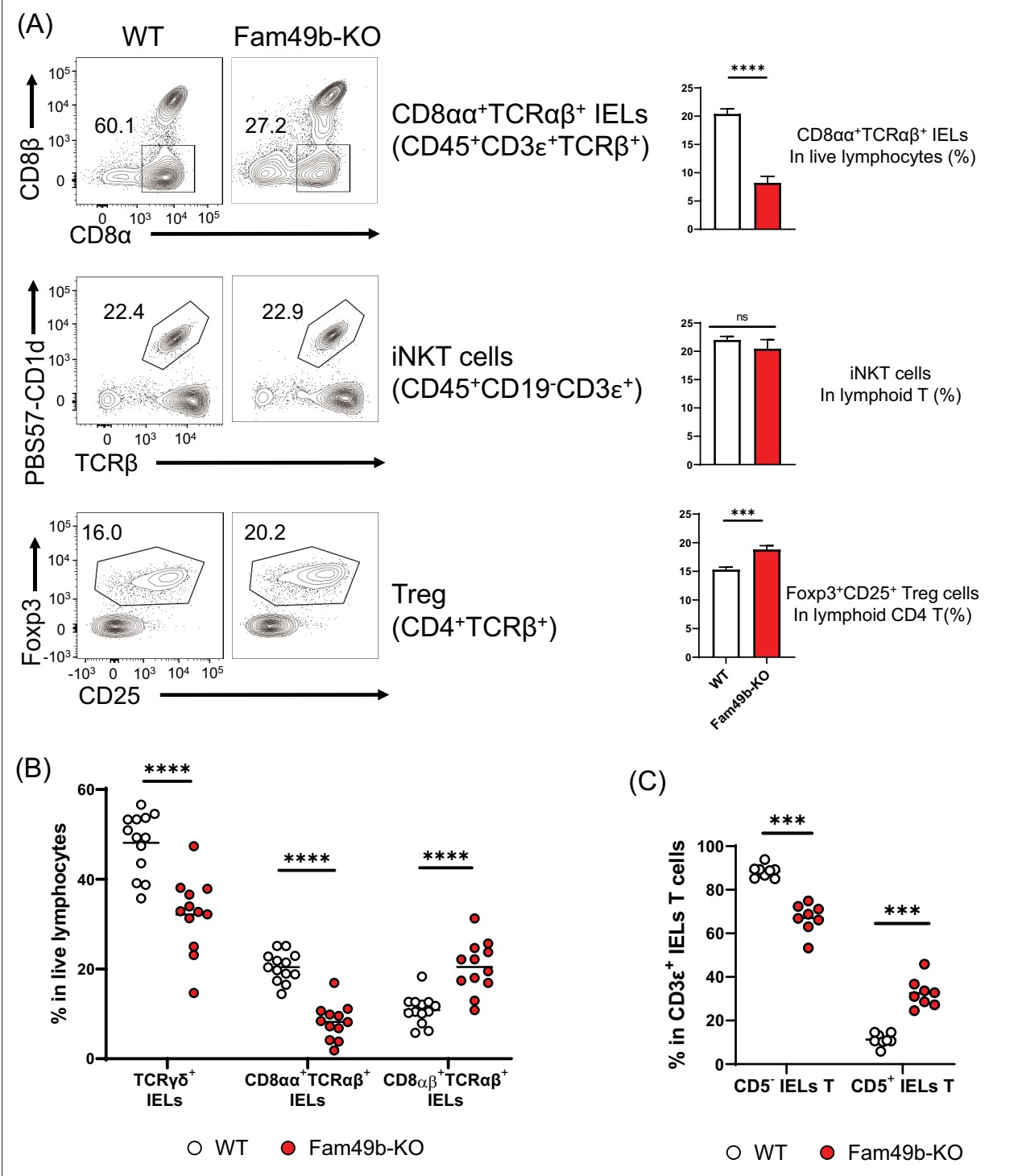

**Figure 7.** Family with sequence similarity 49 member B (Fam49b)-KO mice have lower frequency of CD8αα⁺TCRαβ⁺ and TCRγδ⁺ IELs T cells than wild-type (WT) mice. (**A**) Flow cytometry analysis of CD8αα⁺ TCRβ⁺ IELs T cells (**top**), CD1d-tetramer⁺ iNKT cells in the liver (**middle**), and Foxp3⁺CD25⁺ lymphoid regulatory T cells in the peripheral lymph nodes (**bottom**) from WT and Fam49b-KO mice. Right panels show average frequencies of each population among total lymphocytes or CD4 T cells. ***p=0.0003 and ****p<0.0001 (Mann-Whitney test). Data are pooled from seven independent experiments (CD8αα⁺ TCRβ⁺ IELs; mean and SEM, n=12–13), representative of four experiments (iNKT cells; mean and SEM, n=6), or representative from seven independent experiments (Treg; mean and SEM, n=8). See also *Figure 7—source data 1*. (**B**) Frequency of TCRγδ⁺ IELs T cells, CD8αα⁺TCRαβ⁺ IELs T cells, and CD8αβ⁺TCRβ⁺ IELs T cells among total live IELs cells in WT and Fam49b-KO mice. Each dot represents an individual mouse. Small horizontal lines indicate the mean of 12–13 mice. ****p<0.0001 (Mann-Whitney test). Data are pooled from seven independent experiments. See also *Figure 7—source data 1*. (**C**) Frequency of CD5⁺ T cells and CD5⁻ T cells among total CD3ε⁺ IELs T cells in WT and Fam49b-KO

*Figure 7 continued on next page*

*Figure 7 continued*

mice. Each dot represents an individual mouse. Small horizontal lines indicate the mean of 8 mice. ***p=0.0002 (Mann-Whitney test). Data are pooled from six independent experiments. See also *Figure 7—source data 1*.

The online version of this article includes the following source data and figure supplement(s) for figure 7:

**Source data 1.** The numerical data used to generate the *Figure 7*.

**Figure supplement 1.** Flow cytometry gating strategies to analyze IELs T cells.

**Figure supplement 2.** Total number of Treg cells in lymph nodes, and minor IELs T subsets.

**Figure supplement 2—source data 1.** The numerical data used to generate the *Figure 7—figure supplement 2*.

**Figure supplement 3.** Intraepithelial lymphocyte (IEL) precursors in thymus.

**Figure supplement 3—source data 1.** The numerical data used to generate the *Figure 7—figure supplement 3*.

molecule, suggests that the impaired T cell development in Fam49b-KO mice is likely a result of enhanced Rac activity in DP thymocytes.

How might enhanced Rac activity lead to the defective T cell development in Fam49b-KO mice? Rac is known to regulate actin reorganization in T cells through binding with the Rac downstream effectors, such as the PAK and WAVE2 complex (*Kumari et al., 2014*). The Pak2-deficient CD4 thymocytes showed weakened TCR-signaling strength as indicated by the reduction of Nur77 expression in response to αCD3-stimulation (*Phee et al., 2014*), suggesting that the Rac-driven cytoskeleton remodeling is important for downstream events of TCR signaling. Negatively regulated Rac-driven cytoskeleton remodeling could attenuate protrusion and migration process in T cells. Fam49b-deficient cells showed increased cellular spread and reduced protrusion-retraction dynamics (*Fort et al., 2018*; *Yuki et al., 2019*). Moreover, negative selection occurs via lengthy interactions between T cells and APCs, whereas positive selection is transient interactions (*Melichar et al., 2013*). Therefore, it is possible that altered cytoskeleton remodeling activity in Fam49b-KO thymocytes contributed to their elevated TCR-signaling strength and enhanced negative selection, perhaps by prolonging interactions with thymic APCs.

We found that Fam49b-KO mice showed impaired survival in immature and semi-mature SP thymocytes, and peripheral naïve T cells. Activated Rac1 has been shown to phosphorylate Jun N-terminal kinase (JNK), which in turn phosphorylates Jun and leads to the appearance of active AP-1 (*Gaud et al., 2018*; *Wu et al., 2008*). AP-1 is a transcription factor that participates in tuning on transcription of many genes important for T cell activation (*Lee et al., 2018*). Although JNK is required for TCR signaling, JNK can also promote apoptotic signaling by the upregulation of pro-apoptotic genes such as BIM (*Dhanasekaran and Reddy, 2008*). JNK-deficient thymocytes were resistant to cell death induction caused by apoptotic stimuli such as anti-CD3ε, anti-FAS, and TNF-α (*Rincón et al., 1998*; *Behrens et al., 2001*; *Sabapathy et al., 2001*), suggesting that JNK signaling is required for TCR-mediated apoptosis of thymocytes. Moreover, Rac1-mediated Bcl-2 induction is also important for the differentiation of CD4 SP from DP thymocyte lines by preventing TCRβ-induced apoptosis (*Oda et al., 2007*). We suggest, therefore, that Fam49b protein is a fine regulatory protein of TCR signaling to induce thymic development and T cell activation while suppressing apoptosis caused by excessive Rac1/JNK signaling.

Among all the unusual phenotypes of peripheral T cells in Fam49b-KO mice, one surprising yet interesting observation was the significant loss of CD8αα⁺TCRαβ⁺ and TCRγδ⁺ IELs T cells. These T cell subsets were previously defined as unconventional T cells derived from self-reactive thymocytes that mature through agonist selection. The development of agonist-selected T cells relies on relatively strong and sustained TCR signaling which correlates with the magnitude of store-operated Ca2⁺ entry and NFAT activity (*Stritesky et al., 2012*; *Oh-Hora et al., 2013*). Yet it remains unclear why these cells that receive unusually high TCR signal are not eliminated through negative selection, but instead traffic into the gut and become IEL T cells (*Ruscher et al., 2017*; *Pobezinsky et al., 2012*). Interestingly, thymocytes undergoing agonist selection into CD8αα⁺ TCRαβ⁺ IELs T cells exhibited a rapid and confined migration pattern, in contrast to negatively selecting cells, which showed arrested migration (*Kurd et al., 2021*). Fam49b-deficient cells showed increased cellular mobility (*Yuki et al., 2019*). It is tempting to speculate that the overactivation of Rac-1 in Fam49b-KO mice might rescue IEL precursors from negative selection, perhaps by favoring confined migration over migratory arrest after encountering with agonist ligands.

In conclusion, Fam49b is critical for the thymic development of conventional T cells as well as unconventional natural IELs T cells. Interestingly, the function of Fam49b is restrained to the late-phase T cell development, where it dampens TCR signals to avert negative selection of DP thymocytes. The action of Fam49b is key in distinguishing positive from negative selection in thymic development. In addition, Fam49b is essential for the survival of naïve T cells in the peripheral. Thus, our study offers new insights on the association between modulation of TCR-signaling strength, cytoskeleton remodeling, thymic development processes, and peripheral T cells survival.

# Materials and methods

**Key resources table**

| Reagent type (species) or resource | Designation | Source or reference | Identifiers | Additional information |
|---|---|---|---|---|
| Strain, strain background (*Mus musculus*) | C57BL/6 J | Jackson Laboratory | Stock No. 000664, RRID:IMSR_JAX:000664 | |
| Strain, strain background (*Mus musculus*) | CD45.1 | Jackson Laboratory | Stock No. 002014, RRID:IMSR_JAX:002014 | |
| Strain, strain background (*Mus musculus*) | *Cyria*-/- (C57BL/6 J background) | Generation of knockout mice in this paper | | |
| Strain, strain background (*Mus musculus*) | *Cyrib*-/- (C57BL/6 J background) | Generation of knockout mice in this paper | | |
| Antibody | Anti-human Fam49a (Rabbit polyclonal) | Millipore | Cat# SAB1103179 RRID:AB_10900801 | WB (1:1000) |
| Antibody | Anti-Fam49b (Mouse monoclonal, Clone D-8) | Santa Cruz | Cat# SC-390478 | WB (1:2000) |
| Antibody | Anti-GAPDH (Rabbit polyclonal) | Abcam | Cat# ab9485, RRID:AB_307275 | WB (1:2000) |
| Antibody | anti-PLCγ1 (Rabbit monoclonal, Clone D9H10) | Cell Signaling Technology | Cat# 5690 RRID:AB_10691383 | WB (1:1000) |
| Antibody | anti-p-PLCγ1 (Rabbit polyclonal) | Cell Signaling Technology | Cat# 2821 RRID:AB_330855 | WB (1:1000) |
| Antibody | anti-ZAP-70 (Rabbit monoclonal, Clone 99F2) | Cell Signaling Technology | Cat# 2705 RRID:AB_2273231 | WB (1:1000) |
| Antibody | anti-p-ZAP-70 (Rabbit monoclonal, Clone 65E4) | Cell Signaling Technology | Cat# 2717 RRID:AB_2218658 | WB (1:1000) |
| Antibody | anti-p44/42 MAPK (Erk1/2) (Rabbit polyclonal) | Cell Signaling Technology | Cat# 9102 RRID:AB_330744 | WB (1:1000) |
| Antibody | anti-p-p44/42 MAPK (Erk1/2) (Rabbit monoclonal, Clone 197G2) | Cell Signaling Technology | Cat# 4377 RRID:AB_331775 | WB (1:1000) |
| Antibody | anti-p-LAT (Rabbit monoclonal, Clone E3S5L) | Cell Signaling Technology | Cat# 20172 RRID:AB_3073971 | WB (1:1000) |
| Antibody | anti-LAT (Rabbit monoclonal, Clone E3U6J) | Cell Signaling Technology | Cat# 45533 | WB (1:1000) |
| Antibody | anti-BIM (Rabbit monoclonal, Clone C34C5) | Cell Signaling Technology | Cat# 2933 RRID:AB_1030947 | WB (1:1000) |
| Antibody | anti-Bcl-2 (Rabbit monoclonal, Clone D17C4) | Cell Signaling Technology | Cat# 3498 RRID:AB_1903907 | WB (1:1000) |
| Antibody | anti-p-PAK1 (Ser144)/ PAK2 (Ser141) (Rabbit polyclonal) | Cell Signaling Technology | Cat# 2606 RRID:AB_2299279 | WB (1:1000) |
| Antibody | anti-p-PAK1 (Ser199/204)/ PAK2 (Ser192/197) (Rabbit polyclonal) | Cell Signaling Technology | Cat# 2605 RRID:AB_2160222 | WB (1:1000) |
| Antibody | anti-p-PAK1 (Thr423)/ PAK2 (Thr402) (Rabbit polyclonal) | Cell Signaling Technology | Cat# 2601 RRID:AB_330220 | WB (1:1000) |

*Continued on next page*

*Continued*

| Reagent type (species) or resource | Designation | Source or reference | Identifiers | Additional information |
|---|---|---|---|---|
| Antibody | anti-PAK1 (Rabbit polyclonal) | Cell Signaling Technology | Cat# 2602 RRID:AB_330222 | WB (1:1000) |
| Antibody | anti-PAK2 (Rabbit monoclonal, Clone C17A10) | Cell Signaling Technology | Cat# 2615 RRID:AB_2267950 | WB (1:1000) |
| Antibody | anti-PAK3 (Rabbit polyclonal) | Cell Signaling Technology | Cat# 2609 RRID:AB_2225298 | WB (1:1000) |
| Antibody | anti-PAK1/2/3 (Rabbit polyclonal) | Cell Signaling Technology | Cat# 2604 RRID:AB_2160225 | WB (1:1000) |
| Antibody | Anti-mouse IgG (Dunkey polyclonal) | Licor | Cat# 926–68072, RRID:AB_10953628 | WB (1:10000) |
| Antibody | Anti-Rabbit IgG (Goat polyclonal) | Licor | Cat# 926–32211, RRID:AB_621843 | WB (1:1000) |
| Antibody | Anti-CD3ε-FITC (Rat monoclonal, Clone 17A2) | BioLegend | Cat# 100204, RRID:AB_312661 | FC (1:200) |
| Antibody | Anti-CD4-APC (Rat monoclonal, Clone RM4-5) | BioLegend | Cat# 100516, RRID:AB_312719 | FC (1:200) |
| Antibody | Anti-CD5-APC-Cy7 (Rat monoclonal, Clone 53–7.3) | BioLegend | Cat# 100649, RRID:AB_2860587 | FC (1:200) |
| Antibody | Anti-CD8α-BV421 (Rat monoclonal, Clone 53–6.7) | BioLegend | Cat# 100738, RRID:AB_11204079 | FC (1:200) |
| Antibody | Anti-CD8β-PE-Cy7 (Rat monoclonal, Clone 53–5.8) | BioLegend | Cat# 140416, RRID:AB_2564385 | FC (1:200) |
| Antibody | Anti-CD11b-PerCP-Cy5.5 (Rat monoclonal, Clone M1/70) | BioLegend | Cat# 101228, RRID:AB_893232 | FC (1:200) |
| Antibody | Anti-CD19-PerCP-Cy5.5 (Rat monoclonal, Clone 6D5) | BioLegend | Cat# 115534, RRID:AB_2072925 | FC (1:200) |
| Antibody | Anti-CD24-FITC (Rat monoclonal, Clone M1/69) | BioLegend | Cat# 101806, RRID:AB_312839 | FC (1:200) |
| Antibody | Anti-CD25-APC (Rat monoclonal, Clone PC61) | BioLegend | Cat# 102012, RRID:AB_312861 | FC (1:200) |
| Antibody | Anti-CD44-Pacific Blue (Rat monoclonal, Clone IM7) | BioLegend | Cat# 103020, RRID:AB_493683 | FC (1:200) |
| Antibody | Anti-CD45-Pacific Blue (Rat monoclonal, Clone 30-F11) | BioLegend | Cat# 103126, RRID:AB_493535 | FC (1:200) |
| Antibody | Anti-CD45.1-PE (Mouse monoclonal, Clone A20) | BioLegend | Cat# 110708, RRID:AB_313497 | FC (1:200) |
| Antibody | Anti-CD45.2-FITC (Mouse monoclonal, Clone 104) | BioLegend | Cat# 109806, RRID:AB_313443 | FC (1:200) |
| Antibody | Anti-CD45R/B220-PerCP-Cy5.5 (Rat monoclonal, Clone RA3-6B2) | BioLegend | Cat# 103236, RRID:AB_893354 | FC(1:400) |
| Antibody | Anti-CD62L-APC (Rat monoclonal, Clone MEL-14) | BioLegend | Cat# 104412, RRID:AB_313099 | FC (1:200) |
| Antibody | Anti-CD69-PE (Armenian Hamster monoclonal, Clone H1.2F3) | BioLegend | BioLegend Cat# 104507, RRID:AB_313110 | FC (1:200) |
| Antibody | Anti-CD103-Alexa Fluor 700 (Armenian Hamster monoclonal, Clone 2E7) | BioLegend | BioLegend Cat# 121442, RRID:AB_2813993 | FC (1:200) |
| Antibody | Anti-CD197/CCR7-BV421 (Rat monoclonal, Clone 4B12) | BioLegend | Cat# 120120, RRID:AB_2561446 | FC (1:50) |

*Continued on next page*

*Continued*

| Reagent type (species) or resource | Designation | Source or reference | Identifiers | Additional information |
|---|---|---|---|---|
| Antibody | Anti-Ly6G-PerCP-Cy5.5 (Rat monoclonal, Clone 1A8) | BioLegend | Cat# 127616, RRID:AB_1877271 | FC (1:200) |
| Antibody | Anti-Ly6C-PerCP-Cy5.5 (Rat monoclonal, Clone HK1.4) | BioLegend | Cat# 128012, RRID:AB_1659241 | FC (1:200) |
| Antibody | Anti-NK1.1-PerCP-Cy5.5 (Mouse monoclonal, Clone PK136) | BioLegend | Cat# 108728, RRID:AB_2132705 | FC (1:200) |
| Antibody | Anti-TCRβ-PE/Cy7 (Armenian Hamster monoclonal, Clone H57-597) | BioLegand | Cat# 109222, RRID:AB_893625 | FC (1:200) |
| Antibody | Anti-TCRγδ-PE (Armenian Hamster monoclonal, Clone GL3) | BioLegend | Cat# 118108, RRID:AB_313832 | FC (1:200) |
| Antibody | Anti-CD16/CD32 (Rat monoclonal, Clone 93) | Thermo Fisher Scientific | Cat# 14-0161-85, RRID:AB_467134 | FC (1:200) |
| Antibody | Anti-Cleaved Caspase 3-PE (Rabbit monoclonal, Clone D3E9) | Cell Signaling Technology | Cat# 12768, RRID:AB_2798021 | FC (1:50) |
| Antibody | Anti-Foxp3-PE (Rab monoclonal, Clone FJK16s) | eBioscience | Cat# 12-5773-82, RRID:AB_465936 | FC (1:200) |
| Antibody | Anti-TCRVα2-FITC (Rat monoclonal, Clone H57-597) | BioLegand | Cat# 127806, RRID:AB_1134188 | FC (1:200) |
| Antibody | Anti-IL-7RαBV650 (Rat monoclonal, Clone A7R34) | BioLegand | Cat# 135043, RRID:AB_2629681 | FC (1:200) |
| Antibody | Anti-PD-1 APC (Rat monoclonal, Clone RMP1-30) | BioLegand | Cat# 109112, RRID:AB_10612938 | FC (1:200) |
| Antibody | Anti- LPAM-1 (Integrin α4β7) PE (Rat monoclonal, Clone DATK32) | BioLegand | Cat# 120606, RRID:AB_493267 | FC (1:200) |
| Antibody | Biotin anti-mouse CD3 antibody | Biolegend | Cat# 100244 RRID:AB_2563947 | 60 µg/ml |
| Antibody | Biotin anti-mouse CD4 antibody | Biolegend | Cat# 100404 RRID:AB_312688 | 60 µg/ml |
| Peptide, recombinant protein | Streptavidin | SouthernBiotech | Cat# 7100–01 | 60 µg/ml |
| Sequence-based reagent | *Cyrib*_F | This paper | PCR primers | AGGAGCTGGC CACGAAATAC |
| Sequence-based reagent | *Cyrib*_R | This paper | PCR primers | GGCGTACTAGTC AAGGCTCC |
| Sequence-based reagent | *Actb*_F | This paper | PCR primers | CCTGAACCCTAAG GCCAACC |
| Sequence-based reagent | *Actb*_R | This paper | PCR primers | ATGGCGTGAGG GAGAGCATA |
| Commercial assay or kit | RNeasy Plus Micro Kit | QIAGEN | Cat# 74034 | |
| Commercial assay or kit | SuperScript IV Firs-Strand Synthesis Reaction, | Thermo Fisher Scientific | Cat# 18091050 | |
| Commercial assay or kit | PowerUp SYBR Green Master Mix | Thermo Fisher Scientific | Cat# A25741 | |
| Commercial assay or kit | Liver dissociation kit, mouse | MACS | Cat# 130-105-807 | |
| Chemical compound, drug | DTT | Fisher Scientific | Cat# BP172-5 | (1 mM) |
| Software, algorithm | Image Studio Lite | | RRID:SCR_013715 | v5.2.5 |
| Software, algorithm | FlowJo | FlowJo | RRID:SCR_008520 | v10.7.1 |
| Software, algorithm | GraphPad Prism | GraphPad Software | RRID:SCR_002798 | v9 |

| Reagent type (species) or resource | Designation | Source or reference | Identifiers | Additional information |
|---|---|---|---|---|
| Other | Ghost Dye Violet 510 Viability dye | Tonbo Bioscience | Cat# 13–0870 T100 | FC (1:1000) |

## Mice

C57BL/6 J (WT, stock no: 000664) and CD45.1 mice (B6.SJL-Ptprca Pepcb/BoyJ, stock no: 002014) were purchased from the Jackson Laboratory and bred in-house. Fam49a-KO and Fam49b-KO mice were generated by CRISPR/Cas9 gene-editing technology. The construct was electroporated into embryonic stem cells at the University of California at Berkeley gene targeting facility. All mouse procedures were approved by the Johns Hopkins University Animal Care and Use Committee and were following relevant ethical regulations (M021M261).

## Antibodies and reagents

Western blotting: anti-Fam49a (1103179) Millipore Sigma (St. Louis, MO); anti-Fam49b (D-8) Santa Cruz (Dallas, Texas); anti-GAPDH (ab9485) Abcam (Waltham, MA); anti-PLCγ1 (D9H10), anti-p-PLCγ1, anti-ZAP-70 (99F2), anti-p-ZAP-70, anti-ERK 1/2, anti-p-ERK 1/2 (197G2), anti-LAT (E3U6J), anti-p-LAT (E3S5L), anti-BIM (C34C5), anti-Bcl-2 (D17C4), anti-p-PAK 1(Ser144)/2 (Ser141), anti-p-PAK 1(Ser199/204)/2(Ser192/197), anti-p-PAK 1(Thr423)/PAK 2(Thr402), anti-PAK 1, anti-PAK 2 (C17A10), anti-PAK-3, anti-PAK 1/2/3, Cell Signaling Technology (Danvers, MA); anti-mouse IgG (926–68072), anti-rabbit IgG (926–32211) Li-cor (Lincoln, NE). - Flow cytometry: anti-CD3e (17A2), anti-CD4 (RM4-5), anti-CD5 (53–7.3), anti-CD8α (53–6.7), anti-CD8β (53–5.8), CD11b (M1/70), anti-CD19 (ID3), anti-CD24 (M1/69), anti-CD25 (PC61), anti-CD44 (IM7), anti-CD45 (30-F11), anti-CD45.1 (A20), anti-CD45.2 (104), anti-CD45R/B220 (RA3-6B2), anti-CD62L (MEL-14), anti-CD69 (H1.2F3), anti-CD197/CCR7 (4B12), anti-Ly6G (IA8), anti-Ly-6C (HK1.4), anti-NK1.1 (PK136), anti-TCRβ (H57-597), anti-TCRγδ (GL3), anti-TCRVα2 (B20.1), anti-IL-7Rα (A7R34), anti-Integrin α4β7 (DATK32), anti-CD103 (2E7) anti-PD-1 (RMP1-30), BioLegend (San Diego, CA); anti-CD16/32 (93) ThermoFisher Scientific Waltham, MA; anti-Cleaved Caspase 3 (D3E9) Cell signaling (Danvers, MA); Foxp3 (FJK-16s) eBioscience (San Diego, CA).Tetramerization: PBS-57 loaded mouse CD1d monomers were synthesized by the Tetramer Core Facility of the US National Institute of Health, Streptavidin-APC (PJ27S) and Streptavidin-RPE (PJRS27-1) were purchased from Prozyme (Agilent, Santa Clara, CA).

## T cell activation assay

Total thymocytes were rested for 4 hr in RPMI medium at 37°C, 5%. After resting, cells were incubated with biotin-conjugated anti-CD3 (60 µg/ml, 145–2 C11) and anti-CD4 (60 µg/ml, GK1.5) for 20 min on ice. Cells were washed and incubated for 5 min on ice with streptavidin (60 µg/ml, SouthernBiotech, USA) for cross-linking and incubated for the indicated time at 37°C. 1 ml of cold PBS was added at the end of stimulation and cell pellets were lysed for western blotting.

## Immunoblotting

Cell extracts were prepared by resuspending cells in PBS, then lysing them in RIPA buffer containing protease inhibitor cocktail (Thermo Fisher Scientific, Waltham, MA). Protein concentrations were determined with the BCA Protein Reagent Kit (Pierce, ThermoFisher Scientific), after which 2-mercaptoethanol and 4 x Laemmli Sample buffer (Bio-Rad, Hercules, CA) were added and the samples were boiled. Western blotting was performed according to standard protocols using anti-Fam49a pAb, and anti-Fam49b mAb, and anti-GAPDH pAb. IRDye800CW conjugated goat anti-rabbit and IRDye680RD conjugated donkey anti-mouse were used as secondary antibodies. The membrane was scanned with the Odyssey Infrared Imaging System (Li-cor, model 9120).

## Real-time RT-PCR

The subsets of C57BL/6 thymocytes was collected using BD FACSAria II Cell sorter by the Ross Flow Cytometry Core Facility of the johns Hopkins. DN1 cells were gated as CD25⁻CD44^hi; DN2 cells were gated as CD25⁺CD44^int–hi; DN3 cells were gated as CD25⁺CD44^neg–lo; and DN4 cells were gated as CD25⁻CD44⁻. Total RNA was isolated using the RNeasy Plus Micro Kit (Qiagen, Germantown, MD) and cDNA was amplified by SuperScript IV First-Strand Synthesis (Invitrogen, ThermoFisher Scientific) according to the manufacturer's instructions. Real-time PCR was performed using SYBRgreen

PCR Master Mix (Applied Biosystems, ThermoFisher Scientific) and the ViiA 7 Real-Time PCR System (Applied Biosystems, Thermo Fisher Scientific). Fam49b primers were forward, 5'-AGGAGCTGGCCA CGAAATAC-3', and reverse, 5'- GGCGTACTAGTCAAGGCTCC-3'. Results were normalized to β-actin expression with the 2−ΔCt method.

## Isolation of immune cells

Small-intestine IELs were isolated as previously described (*Qiu and Sheridan, 2018*): Changes were made as follows: DTT (BP172-5, Fisher Scientific) was used instead of DTE. Immune cells were collected from the interface of the 44% and 67% Percoll gradient and characterized by flow cytometry. Hepatic lymphocytes were isolated using a Liver dissociation kit (Miltenyi Biotec, Auburn, CA) according to the manufacturer's instructions. Samples were resuspended in 33% Percoll and spun, and the cell pellet was collected and labeled for flow cytometry. Naïve CD4$^+$ T cell in the spleen and lymph nodes from C57BL/6 or FAM49b-KO mice were sorted using Naïve CD4$^+$ T cell Isolation Kit (Miltenyibiotec, USA) according to the manufacturer instructions. Samples were resuspended in RIPA buffer for western blot.

## Generation of bone marrow chimera

T cell-depleted bone marrow cells from CD45.2$^+$ C57BL/6, Fam49a-KO mice, or Fam49b-KO mice (1×10$^6$ cells) were used to reconstitute sublethally irradiated (1000 rad) CD45.1$^+$ wild-type mice by i.v. injection. Reconstituted mice were analyzed 8 weeks after bone marrow transfer.

## Cell staining

For cleaved caspase 3 staining (*Breed et al., 2019*), homogenized mice thymocyte cells were stained with anti-CCR7/CD197 at a final dilution of 1:50 for 30 min at 37 °C prior to additional surface stains. Following surface staining, cells were fixed with Cytofix/Cytoperm (BD Biosciences) for 20 min at 4 °C. Cells were then washed with Perm/Wash buffer (BD Biosciences) twice. Cells were stained with anti–cleaved caspase 3 at a 1:50 dilution at 23 °C for 30 min.

For iNKT staining, Biotinylated PBS-57 loaded or unloaded monomers were obtained from the Tetramer Core Facility of the National Institutes of Health and tetramerized with PE-labeled strepta-vidin from ProZyme. Hepatic lymphocytes were resuspended in 100 μl of sorter buffer (PBS with 2% FCS, 1 mM EDTA, and 0.1% sodium azide) and stained with PE-iNKT tetramers at a final dilution of 1:200 at 23 °C for 30 min.

For transcription factor staining, cells were incubated with surface antibody at 4 °C for 20 min, permeabilized at 4 °C for 30 min, and then stained with anti–Foxp3 at 23 °C for 30 min using a Foxp3/ Transcription transcription factor buffer set (Invitrogen, ThermoFisher Scientific).

For 7AAD/Annexin V staining, cells were incubated with surface antibody at 4 °C for 20 min and then stained with anti–7-AAD and Annexin V at 23 °C for 15 min using an Annexin V apoptosis detec-tion kit with 7-AAD (Biolegend). Samples were acquired with BD FACSCelesta (BD Biosciences), and data were analyzed with FlowJo (version 10.7.1).

## Statistical testing

GraphPad Prism was used for all statistical analyses. A nonparametric Mann-Whitney U test or one-way ANOVA was used for the estimation of statistical significance. Data is shown as mean ± SEM. *p<0.05, **p<0.01, ***p<0.001, ****p<0.0001.

## Acknowledgements

We thank Dr. J David Peske for his editorial comments. Dr. Nilabh Shastri passed away in 2021. May he rest in peace. This work was supported by NIH grant (R01AI130210, R01AI121174, R37AI060040) and the Korea government (MSIT) (RS-2023–00212022, MRC-2017R1A5A2015541).

## Additional information

### Funding

| Funder | Grant reference number | Author |
| --- | --- | --- |
| National Institutes of Health | R01AI130210 | Scheherazade Sadegh-Nasseri |
| National Institutes of Health | R01AI121174 | Scheherazade Sadegh-Nasseri |
| National Institutes of Health | R37AI060040 | Scheherazade Sadegh-Nasseri |
| Ministry of Science and ICT, South Korea | RS-2023-00212022 | Chan-Su Park |
| Ministry of Science and ICT, South Korea | MRC-2017R1A5A2015541 | Chan-Su Park |

The funders had no role in study design, data collection and interpretation, or the decision to submit the work for publication.

### Author contributions

Chan-Su Park, Conceptualization, Investigation, Visualization, Methodology, Writing - original draft, Project administration, Writing – review and editing; Jian Guan, Investigation, Writing – review and editing; Peter Rhee, Hee-sung Lee, Ji-hyun Park, Investigation; Federico Gonzalez, Resources; Laurent Coscoy, Ellen A Robey, Supervision, Writing – review and editing; Nilabh Shastri, Supervision, Funding acquisition; Scheherazade Sadegh-Nasseri, Supervision, Funding acquisition, Writing – review and editing

### Author ORCIDs

Chan-Su Park (ID) https://orcid.org/0000-0003-4968-8304
Jian Guan (ID) https://orcid.org/0000-0002-0118-6578
Laurent Coscoy (ID) https://orcid.org/0000-0002-7337-2345
Ellen A Robey (ID) https://orcid.org/0000-0002-3630-5266
Nilabh Shastri (ID) https://orcid.org/0000-0002-8060-3025

### Ethics

All mouse procedures were approved by the Johns Hopkins University Animal Care and Use Committee and were following relevant ethical regulations (M021M261).

### Decision letter and Author response

Decision letter https://doi.org/10.7554/eLife.76940.sa1
Author response https://doi.org/10.7554/eLife.76940.sa2

---

## Additional files

### Supplementary files
• Transparent reporting form

### Data availability

All data generated or analysed during this study are included in the manuscript and supporting files. Source data files has been provided for Figures 2–7 and their accompanying figure supplements. Sanger Sequencing data for Cyria and Cyrib, and immunoblot for Cyria (Fam49a) and Cyrib (Fam49b) have been provided for Figure 1 (Figure 1—source data 1 and 2, respectively).

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
