## [Editor Report]

The protein called Family with sequence similarity 49 member B (in short Fam49b) is a newly discovered negative regulator of TCR signaling that suppresses Rac-1 activity in vitro. The present manuscript analyzes in a comprehensive and solid manner the role of Fam49b during thymic T cell development. It is a very valuable piece of work in that it demonstrates that the Fam49b protein dampens thymocyte TCR signaling allowing thymocytes to escape negative selection.

---

## [Decision Letter]

**Decision letter after peer review:**

Thank you for submitting your article "Fam49b dampens TCR signal strength to regulate survival of positively selected thymocytes" for consideration by *eLife*. Your article has been reviewed by 3 peer reviewers, and the evaluation has been overseen by a Reviewing Editor and Tadatsugu Taniguchi as the Senior Editor. The reviewers have opted to remain anonymous.

Essential revisions (for the authors):

The three Reviewers are convergent views and consider that the experiments are well performed and the results are convincing. They consider that

1) having data from TCR transgenic mice,

2) adding some data from experiments in response to the questions that were raised, would greatly improve the manuscript.

*Reviewer #1 (Recommendations for the authors):*

1. Thymocyte selection (and the effects of mutations on selection) is difficult to explore in mice with a polyclonal T cell repertoire, but can be studied in more detail in TCR transgenic (TCR-tg) mice that express a TCR with known specificity. It would be informative to see the phenotype of TCR-tg x Fam49b mice. Any possible effects on positive selection, in particular, would be better revealed with this model and the proposed shift from positive to negative selection could be better demonstrated. Also, CD28 has been shown to be critical for negative selection. Analysis of CD28 KO/Fam49b KO mice would be very informative as it would be predicted that the assumed negative selection would be prevented or that there would be a strong diversion of cells to the DN TCR+ thymocyte CD8aa TCR+ iIEL precursor lineage as shown in PMID: 22544394.

2. The deletion strategy shown in Figure 1a,b could result in expression of Fam49a and Fam49b proteins with c-terminal truncations that might be partially active. What epitopes were the antibodies used for Figure 1c raised against? Are these epitopes present in potential truncated proteins expressed in the KO mice? Analysis of mRNA expression by rt-PCR should also be performed.

3. It is possible that Fam49a and Fam49b have partially redundant functions. Did the authors generate double KO mice to analyze if this is feasible?

4. Signaling experiments should be performed with Fam49b thymocytes/T cells to document the predicted effects on signal transduction. A survey of TCR signaling responses is warranted in this study. Analysis of thymocyte adhesion similar to experiments done with Rac-tg mice (PMID: 22544394) would be informative.

*Reviewer #2 (Recommendations for the authors):*

This study is very well designed, but, while there is support for the main message of this study (Fam49b is involved in thymic selection), there are several aspects that should be clarified by performing new experiments in order to improve the manuscript. The authors should consider the following major points:

– One major drawback of this manuscript is the lack of TCR transgenic mice that could have been included to avoid potential bias due to a shift in TCR repertoire. In addition, the authors claim "Our results demonstrate that Fam49b dampens thymocytes TCR signaling in order to escape negative selection …." but there is no single experiment addressing the impact of Fam49b deficiency on TCR signaling in thymocytes. The authors should cross their Fam49b KO mice with TCR Tg in order to analyze if Fam49b deficiency affects calcium flux in a way that is representative of decreased positive selection or enhanced negative selection. Since this protein has been shown to regulate the actin cytoskeleton through Rac1-Pak, we could have expected some signaling studies of this axis and more generally of other pathways (ZAP, Vav1, ERK, Rela, Akt, …) and imaging studies of synapse organization.

– The effect of Fam49b deficiency is relatively mild in the thymus as compared to that in the periphery. The authors should explain why peripheral T cells are so strongly decreased since it is difficult to justify this strong decrease of peripheral T cells (particularly CD8^+^ T cells) simply by increased negative selection.

– Absolute numbers of thymic cells in Figure 2D should be included.

– The authors performed bone marrow chimera experiments to show T-cell intrinsic defects. In Figure 3A, they show the data concerning peripheral T cells but no data about the effect in the thymus despite this sentence: "The Fam49b-KO thymocytes developed in WT thymic environment are like those developed in the germline Fam49b-KO environment in terms of both thymocyte and peripheral lymphocyte phenotypes. Therefore, the effect of Fam49b mutation on T cell development is predominantly due to thymocyte intrinsic functions." This data should be added to the figure.

– The data presented in figure 3C and Figure 3 E show a "block" at the transition from TCRbintCD69+ to TCRbhiCD69+ and TCRbintCD5hi to TCRbhiCD5hi, suggesting defective signals in TCRbintCD69+ and TCRbintCD5hi which leads to inefficient transition. Both TCRbintCD69+ and TCRbintCD5hi are still very enriched with DP cells (see supplementary data). So, I wouldn't necessarily conclude that post-positive selection events are affected by this deficiency. This point should be clarified.

– In Figure 4B, the staining of CD69 in CD8^+^ T cells is weird and is suggestive that CD3low cells were not excluded in the gating strategy. Did the authors exclude CD3 low CD8^+^ T cells from their analysis on CD8SP thymocytes in general?

It is also difficult to conclude that the increased CD5 expression on DP cells is due to higher TCR signaling or rather to a shift in TCR repertoire towards higher affinity TCR. An analysis in TCR transgenic mice would resolve this issue

– In Figure 4C, the IL-2 production was tested after three days of stimulation. This is too late since the peak of IL-2 production occurs in general 24h after stimulation. Kinetics of IL-2 production will be more informative.

– The data presented in Figure 5 concerning the differential impact of Fam49b deficiency on CD8aa TCRab+ IELs, liver iNKT cells and lymph node Tregs Tregs are very descriptive, without any data on the development in the thymus. This should be clarified.

*Reviewer #3 (Recommendations for the authors):*

1) Figure 1: it would be informative to add the sequencing trace files for the confirmation of the mutation. Furthermore, after which amino acid (what is the protein length of Fam49a and Fam49b) is the stop codon? Is it possible that a truncated form of Fam49a and Fam49b is expressed? Would the antibodies that have been used for the western blots detect truncated forms? Based on the information provided, it is not clear whether the antibodies against Fam94a and Fem49b recognize the N, C terminus or another region of the proteins.

2) Figure 2A and B: It would be informative to also show the percentages and numbers of TCRb+ cells as well as to pre-gate on TCRb+ cells when displaying CD4 and CD8a expression. Is there an increase in DN cells in the periphery? Which marker was used for B cells? CD19? This is not indicated.

3) In Figure panel 2D: it would be important to pre-gate on TCRb-high cells for determining the CD4/CD8SP ratio. Otherwise, the CD8SP population includes the CD8 ISP subset.

4) Figure 3: Did the authors also analyze CD44 and CD62L expression in the mixed BM chimeric mice? This would help to understand whether the altered CD44/CD62L expression (relative reduction of the frequency of naive cells) shown in Figure 2C is T cell-intrinsic or potentially a consequence of (a partial) T cell homeostatic expansion due to T cell lymphopenia.

5) The authors conclude that the reduced numbers of SP cells and peripheral T cells is due to a defect in negative selection. How can the authors exclude that the reduced numbers of T cells is not due to a general survival defect of post positive selection SP thymocytes? What about expression of pro-apoptotic and pro-survival factors such as Bim and Bcl2, respectively? And what about IL-7 receptor α expression, which is upregulated during positive selection. The expression of these factors should be analyzed.

6) The conclusion that negative selection is altered is based on the results shown in Figure 4A. The frequency of cleaved Caspase-3+ cells is very low. To strengthen the conclusion it would be good to include other approaches that indicate altered negative selection. What happens if WT and Fam49b KO thymocytes are cultured overnight in the presence/absence of anti-CD3 stimulation: would this also reveal changes in cleaved Caspase3? What about superantigen (SEB)-induced deletion of specific TCR Vb isoforms? This is another readout that can be used to study negative selection.

7) How does altered signaling strength in Fam49b KO thymocytes affect the development of thymic Foxp3+ Tregs and thymic NKT cells? Is there a difference in the frequency of these cells in the thymus? In Figure 5, data are shown from LN (Foxp3) and liver (iNKT) but no data are shown from the thymus.

8) The data from peripheral T cells are not convincingly supporting the conclusion that TCR signaling strength is altered. Cell proliferation is normal, wouldn't one expect changes in proliferation if signaling strength in KO cells is changed? What about the upregulation of activation markers such as CD69 and CD25? It would also be informative to plot Cell trace violet dilution with intracellular IL-2 staining to determine whether enhanced IL-2 is linked with a particular division stage. What about other cytokines, e.g. TNF or IFNγ? And what happens if cells are stimulated with anti-CD3/CD28 – would there still be a difference in IL-2?

9) The IEL data are interesting. It would be good to show the gating strategy as well as a representative contour plot for all subsets shown in the diagram in panel 5B and 5C. However, the reduction of the natural IELs could be due to changes in negative selection or – similar to my comment #5 – reduced survival of selected cells (without affecting negative selection). This should be at least discussed as an alternative explanation.

[Editors' note: further revisions were suggested prior to acceptance, as described below.]

Thank you for resubmitting your work entitled "Fam49b dampens TCR signal strength to regulate survival of positively selected thymocytes" for further consideration by *eLife*. Your revised article has been evaluated by Tadatsugu Taniguchi (Senior Editor) and a Reviewing Editor.

The 3 Reviewers appreciate the important additions contained in the revised manuscript such as the inclusion of data from TCR transgenic mice that improve the quality and impact of the study. However and as outlined below by the 3 Reviewers several precise issues need to be addressed prior to publication in *eLife*.

*Reviewer #1 (Recommendations for the authors):*

The revised manuscript contains some important additions such as the inclusion of data from TCR transgenic mice that improve the quality and impact of the study. That said, this report is mainly descriptive with very little mechanistic insight. It was hoped that the requested signaling/biochemistry experiments would contribute to mechanism, but as stated below, the results of these studies are not trustworthy. Several issues need to be addressed prior to publication:

1. The observation that Fam49a KO mice have no obvious phenotype is not a valid reason for not generating Fam49a/49b double KO (DKO) mice. As mentioned in the review, there could be partial redundancy between Fam49a and 49b and the DKO could have a phenotype more severe than the Fam49b KO. This reviewer does not insist on the generation of the DKO (though it is unfortunate) but does want to make the point that the authors' reasoning for not generating the DKO is flawed.

2. The CD69 surface expression results are perplexing. While CD5 surface expression is "analog" and parallels TCR signal intensity, CD69 expression has historically been found to be more "digital" (i.e., CD69+ and CD69- populations rather than a shift in MFI), though the data in Figure 4b indicate an analog pattern (the differences shown are also very slight).

3. In Figure 5a and 5b, how do the authors explain normal numbers of semi-mature cells when the numbers of immature and mature cells are reduced?

4. The results in Figure 5c are not convincing (that there is a difference in the expression of IL-7ra).

5. In Figure 5d,e how do the authors explain increased cell death of semi-mature cells when these cells were not reduced (Figure 5a,b)?

6. Higher expression of CD5 and CD69 in Figure 6 is not convincing.

7. Figure 6d, how do the authors explain the low % of naive OTI T cells in the control mice? Typically, >90% of OTI CD8^+^ T cells are naive in lympho-replete mice.

8. Figure 6f,g, these results are the most problematic in the revised paper. pERK should be virtually undetectable in unstimulated thymocytes and T cells. Also, there is no increase in pERK after stimulation indicating that the cells are not activated. Other effectors (pLAT, pZAP, pZAP-70) should be checked. Several experiments should be performed and statistics shown in Figure 6g. Most notably, the authors do not evaluate Rac activation in the mutant mice to show that it is elevated as predicted!

9. The results in Figure 7 are interesting but the authors did not perform the requested experiment, namely to generate CD28 KO / Fam49b KO (DKO) mice to test if negative selection is indeed increased in Fam49b mice. Deletion of CD28 would allow the rescue of otherwise negatively selected thymocytes, and an increase in DN, TCR+, PD-1+ cells in CD28/Fam49b DKO mice compared to CD28 KO mice would provide strong support for their hypothesis that negative selection is increased in the absence of Fam49b.

10. Supplement 3a and 3b are mentioned in the text but not included in the figures.

*Reviewer #2 (Recommendations for the authors):*

We would like to thank the authors for performing several experiments that resolved some issues raised during my first reviewing process. These new experimental data sets revealed that Fam49b, besides its impact on thymic selection, plays an important role in T cell survival in the periphery. Based on these new findings, the authors should amend the discussion that is mainly focused on thymic selection. In addition, the data presented concerning the impact Fam49b on TCR signaling are not convincing and concerns only ERK without quantification and statistical analyses. This part should be strengthened by analyzing other signaling pathways.

[Editors' note: further revisions were suggested prior to acceptance, as described below.]

Thank you for resubmitting your work entitled "Fam49b dampens TCR signal strength to regulate survival of positively selected thymocytes and peripheral T cells" for further consideration by *eLife*. Your revised article has been evaluated by Tadatsugu Taniguchi (Senior Editor) and a Reviewing Editor.

The manuscript has been improved but there are some remaining issues that need to be addressed, as outlined below:

The authors' explanation for the perplexing results in Figure 5a and 5b (i.e., that proliferation differences could explain why the numbers of semi-mature cells are normal in the KO when the numbers of immature and mature cells are reduced) is unsatisfactory. The reviewer is unaware of any data showing that SP thymocytes proliferate at any stage of their maturation. If I am mistaken, please provide the citation for these findings. I think it more likely that the gating (particularly distinguishing CD62L-lo from -neg and CD69-lo from -neg) is prone to error. In any event, the finding that semi-mature cells are not reduced but immature and mature cells are reduced it very confusing. Because I believe this is a technical problem, I suggest just showing the results for only immature and mature cells to avoid this apparent inconsistency.

In Figure 2c, what are CD62L-CD44- cells? Legend says plots are LN T cells not total LN cells, but Figure 2c shows a large population of CD62L-CD44- cells. To our knowledge, no CD62L-CD44- T cell populations have been previously described. Could it be that the cells in Figure 2c are total LN cells?

Line 163 "Taken together, these results suggest that positive selection remains mostly unaffected by the lack of Fam49b molecule, Fam49b plays a more important role in the later stages of T cell development in the thymus." What "later stages" of development are they referring to? Do they mean survival? Also, the reduction of TCR-hi CD69+ cells (Figure 3c) suggests that late stages of positive selection are affected. I would maintain that "post-selection" means CD69-neg and that any CD69+ cell is undergoing selection.

Line 126 "we concluded that Fam49a is unlikely to 127 play a significant role in T cell development" is obviously misleading since the authors mention in their response that the double Fam49a/49b KO has a more severe phenotype than the Fam49a KO.

Since the authors did make OTI TCR transgenic Fam49b KO mice then it would be very helpful to see analysis of positive selection (e.g., as shown in Figure 3c,e, Figure 4b, Figure 5b) to show if the affects observed with polyclonal mice are observed with TCR transgenic mice.

[Editors' note: further revisions were suggested prior to acceptance, as described below.]

Thank you for resubmitting your work entitled "Fam49b dampens TCR signal strength to regulate survival of positively selected thymocytes and peripheral T cells" for further consideration by *eLife*. Your revised article has been evaluated by Tadatsugu Taniguchi (Senior Editor) and a Reviewing Editor.

The manuscript has been improved but there is a remaining issue that need to be addressed, as outlined below:

Therefore, prior to accepting the manuscript please remove the data and the text corresponding to Figure 2c as requested below by Reviewer #1.

*Reviewer #1 (Recommendations for the authors):*

I appreciate the authors efforts to respond to my concerns/comments. I accept their response to all of the points except for the problem with Figure 2c. It is true that there is a very small percentage of CD44-CD62L- T cells that can be detected in B6 mice, however this percentage is typically <5% of CD4 and CD8 cells. In Figure 2c, 16% of B6 CD4^+^ T cells are CD62L-CD44- and 30% of B6 CD8^+^ T cells are CD62L-CD44-. This is abnormal. Further, CD62L+CD44+ central memory cells are missing from the B6 CD4 subset in Figure 2c but are present in B6 CD8^+^ cells. This is also abnormal. Altogether this means that there is a problem with the FACS experiment used for Figure 2c. If the authors are unable to obtain believable CD44 vs CD62L FACS profiles for their control mice then the results from KO mice cannot be interpreted and Figure 2c should be removed as it is misleading as is.

---

## [Author Response]

Essential revisions (for the authors):The three Reviewers are convergent views and consider that the experiments are well performed and the results are convincing. They consider that1) having data from TCR transgenic mice,

We appreciate this kind and helpful suggestion. We generated Fam49b-KO OT-I mice and analyzed thymocytes and peripheral T cells. These new results are presented in Revised Figure 6 and are described in the Result section (page 10, lines 211-233).

2) adding some data from experiments in response to the questions that were raised, would greatly improve the manuscript.

We have added many new experimental data per reviewers’ suggestions. We hope that our point-by-point responses to the reviewer’s comments are clear and satisfactory.

Reviewer #1 (Recommendations for the authors):1. Thymocyte selection (and the effects of mutations on selection) is difficult to explore in mice with a polyclonal T cell repertoire, but can be studied in more detail in TCR transgenic (TCR-tg) mice that express a TCR with known specificity. It would be informative to see the phenotype of TCR-tg x Fam49b mice. Any possible effects on positive selection, in particular, would be better revealed with this model and the proposed shift from positive to negative selection could be better demonstrated.

We appreciate the reviewer for raising this important question. As suggested, we generated Fam49b-KO OT-I mice, and analyzed thymocytes and peripheral T cells (Revised Figure 6). Based on our observation that enhanced TCR-signaling strength intrinsic to Fam49b-KO DP thymocytes leads to excessive clonal deletion in the cortex and medulla (Figure 4A and Figure 4B), we postulated that enhanced TCR-signal strength of Fam49b-KO TCR-Vα2^+^ DP thymocytes could be diverted into negative selection from positive selection. Even though Fam49b-KO TCR-Vα2^+^ DP showed higher expression of CD5 and CD69 molecules (Revised Figure 6—figure supplement 1A-1B), suggesting Fam49b-KO TCR-Vα2^+^ DP thymocytes had received stronger TCR signaling than the WT TCR-Vα2^+^ DP thymocytes, the frequencies and numbers of CD8 SP thymocyte (Revised Figure 6A-6B) as well as peripheral T cells (Revised Figure 6C-6D) were comparable between WT OT-I and Fam49b-KO OT-I mice of 6 weeks of age. We reasoned that the increased TCR signaling of DP in Fam49b-KO OT-I mice is still below the threshold for inducing apoptosis in DP and SP CD8 T cells in the thymus.

Interestingly, Fam49b-KO OT-I mice showed dramatical reductions in total CD8 T cell (Revised Figure 6C), especially naïve CD8 T cells, (Revised Figure 6D) with age. Based on our observation of the dramatic loss of naïve CD8 T cells in Fam49b-KO OT-I mice with age, we hypothesized that Fam49b is required for peripheral T cells survival. To determine if Fam49b-KO deficiency led to increased apoptosis in naïve CD8 T cells, we assessed apoptosis of peripheral naïve CD8 T cells in Fam49b-KO OT-I mice using 7-AAD and annexin V staining. We found that apoptosis of peripheral T cells was markedly elevated in naïve CD8 T cells from Fam49b-KO OT-I mice (Revised Figure 6E) and polyclonal Fam49b-KO mice (Revised Figure 6—figure supplement 1D). In summary, by analyzing Fam49b-KO OT-I mice, we found new function of Fam49b molecules on peripheral T cells. Fam49b plays an important role in the maintenance and survival of peripheral T cells. These results are presented in Revised Figure 6 and are described in the Result section (page 10, lines 211-233)

Also, CD28 has been shown to be critical for negative selection. Analysis of CD28 KO/Fam49b KO mice would be very informative as it would be predicted that the assumed negative selection would be prevented or that there would be a strong diversion of cells to the DN TCR+ thymocyte CD8aa TCR+ iIEL precursor lineage as shown in PMID: 22544394.

This is an interesting perspective. It has been reported that CD8αα^+^TCRαβ^+^ IELs develop from two distinct CD8αα IEL precursors (IELps) in the thymus, which are Type A IELps (PD-1^+^) and Type B IELps (PD-1^-^) [1]. In that paper, CD28-KO mice, in which self-reactive thymocytes are diverted into the CD8αα IEL lineage, had only more Type A IELps (PD-1^+^), not more Type B IELps (PD-1^−^), than WT mice. We examined the frequencies and numbers of IELps in the thymus from WT and Fam49b-KO mice at 6-7 weeks of age. We observed comparable numbers of Type A IELps, but more Type B IELps in Fam49b-KO mice than in WT mice (Revised Figure 7—figure supplement 3A right panel). These results suggested that Fam49b might be specifically involved in development of thymic Type B IELps lineage, but not Type A IELps lineage, in a CD28-independent manner.

2. The deletion strategy shown in Figure 1a,b could result in expression of Fam49a and Fam49b proteins with c-terminal truncations that might be partially active. What epitopes were the antibodies used for Figure 1c raised against? Are these epitopes present in potential truncated proteins expressed in the KO mice? Analysis of mRNA expression by rt-PCR should also be performed.

We apologize for the lack of clarity. Anti-Fam49a polyclonal antibodies (Σ, SAB 1103179) was produced by synthetic peptide corresponding to amino acids 56-70 of human FAM49a molecules in rabbit. Anti-Fam49b mouse monoclonal antibody (Santa cruz, D-8, sc-390478) binds to 8-20 near the N-terminus of Fam49b of human origin. We clarified this point in the corresponding Figure 1C legend. Fam49a-KO gene ablation is achieved by creating premature stop codon in exon 7, which can be translated into 140 amino acids of truncated Fam49a protein (Revised Figure 1B). Fam49b-KO gene ablation is achieved by creating premature stop codon in exon 6, which can be translated into 117 amino acids of truncated Fam49a protein (Revised Figure 1A). Therefore, potential c-terminal truncated Fam49a or Fam49b proteins could be recognized by Anti-Fam49a Ab or Anti-Fam49b antibody respectively, if the truncated Fam49a or Fam49b proteins are stably expressed in KO mice. However, we did not detect the band expected to be a truncated Fam49a or Fam49b protein by western blotting (Figure 1C). We believed that the potential truncated Fam49a or Fam49b protein can’t be translated or is very unstably expressed in KO mice.

3. It is possible that Fam49a and Fam49b have partially redundant functions. Did the authors generate double KO mice to analyze if this is feasible?

That is an interesting query. However, we did not generate double KO mice because there was no significant difference between WT mice and Fam49a-KO in terms of the phenotype of peripheral T cells and thymocytes subsets. Fam49a-KO mice phenotype results are presented in Figure 2 and are described in the Result section (page 6-7, lines 103-125)

4. Signaling experiments should be performed with Fam49b thymocytes/T cells to document the predicted effects on signal transduction. A survey of TCR signaling responses is warranted in this study. Analysis of thymocyte adhesion similar to experiments done with Rac-tg mice (PMID: 22544394) would be informative.

As suggested, we performed signaling experiments using Fam49b-KO OT-I mice (Revised Figure 6F). We examined activation of ERK because Fam49b is known to negatively regulate Rac-PAK-ERK in T cells [2]. ERK phosphorylation was markedly increased in Fam49b-KO thymocytes than in WT thymocytes, which may support the idea of enhanced TCR signaling in Fam49b-KO thymocytes. We also observed slight increase of BIM/BCl^-^2 ratio in Fam49b-KO naïve OT-I T cells than WT naïve OT-I cells (Revised Figure 6G), suggesting that Fam49b is required for peripheral T cells survival. The corresponding data are presented in Revised Figure 6. The question of the effect of Fam49b on thymocyte adhesion is interesting. Fam49b-KO OT-I mice showed normal T cell development in the thymus, suggesting that Fam49b-KO OT-I thymocytes might not be suitable for thymocyte adhesion experiments.

Reviewer #2 (Recommendations for the authors):This study is very well designed, but, while there is support for the main message of this study (Fam49b is involved in thymic selection), there are several aspects that should be clarified by performing new experiments in order to improve the manuscript. The authors should consider the following major points:– One major drawback of this manuscript is the lack of TCR transgenic mice that could have been included to avoid potential bias due to a shift in TCR repertoire. In addition, the authors claim "Our results demonstrate that Fam49b dampens thymocytes TCR signaling in order to escape negative selection …." but there is no single experiment addressing the impact of Fam49b deficiency on TCR signaling in thymocytes. The authors should cross their Fam49b KO mice with TCR Tg in order to analyze if Fam49b deficiency affects calcium flux in a way that is representative of decreased positive selection or enhanced negative selection. Since this protein has been shown to regulate the actin cytoskeleton through Rac1-Pak, we could have expected some signaling studies of this axis and more generally of other pathways (ZAP, Vav1, ERK, Rela, Akt, …) and imaging studies of synapse organization.

We thank the reviewers for this valuable suggestion. Reviewer #1 also made a similar comment about the issue of Fam49b-KO TCR transgenic mice (Reviewer #1’s comment no. 1. and no. 4). Accordingly, we generated Fam49b-KO OT-I mice, and analyzed thymocytes and peripheral T cells (Revised Figure 6). Unexpectedly, the frequencies and numbers of CD8 SP thymocyte as well as peripheral T cells were comparable between WT and Fam49b-KO OT-I mice of 6 weeks of age (Revised Figure 6A-6D). Therefore, we seriously considered the reviewers’ comments that the effect of Fam49b deficiency is relatively mild in the thymus as compared to that in the periphery (Reviewer #2’s comment no.2) and the decreased number of T cells could have resulted from survival defect (Reviewer #3’s comment no. 5 and no.9). Since we observed significant reduction of naive T cells in Fam49b-KO mice, we postulated that Fam49b deficiency might reduce peripheral T cells survival. As expected, Fam49b-KO OT-I mice showed dramatical reduction of total CD8 T cell (Revised Figure 6C), especially naïve CD8 T cells (Revised Figure 6D) at 12 weeks of age. In agreement with result from 12 weeks old mice, increased apoptosis of naïve T cells was observed in Fam49b-KO OT-I mice (Revised Figure 6E). It has been reported that Fam49b negatively regulate Rac/PAK/ERK/ERK pathway [3-5]. We observed that ERK phosphorylation was dramatically increased in FAM49b-KO OT-I total thymocytes (Revised Figure 6F).

– The effect of Fam49b deficiency is relatively mild in the thymus as compared to that in the periphery. The authors should explain why peripheral T cells are so strongly decreased since it is difficult to justify this strong decrease of peripheral T cells (particularly CD8^+^ T cells) simply by increased negative selection.

We do agree that this is an important question. As mentioned above, we found new function of Fam49b on peripheral T cells. Fam49b plays a key role in maintenance and general survival of peripheral T cells (Revised Figure 6). Moreover, we found that Fam49b deficiency leads to the defective maturation of SP thymocytes (Revised Figure 5). Our findings of defective thymic T cell maturation and peripheral T cell survival in Fam49b-KO mice might explain why the decrease in the number of peripheral T cells is greater than the decrease in the number of SP thymocytes.

– Absolute numbers of thymic cells in Figure 2D should be included.

Per Reviewer’s suggestion, we have added Revised Figure 2E, which shows the total numbers of thymocytes, TCRβ^hi^ CD8 SP, and TCRβ^hi^ CD4 SP. The absolute numbers of TCRβ^hi^ CD4 SP and TCRβ^hi^ CD8 SP were lower in Fam49b-KO mice than WT mice (Revised Figure 2E). This result is consistent with our observation that Fam49b-KO mice have lower numbers of immature SP and mature SP than WT mice (Revised Figure 5B).

– The authors performed bone marrow chimera experiments to show T-cell intrinsic defects. In Figure 3A, they show the data concerning peripheral T cells but no data about the effect in the thymus despite this sentence: "The Fam49b-KO thymocytes developed in WT thymic environment are like those developed in the germline Fam49b-KO environment in terms of both thymocyte and peripheral lymphocyte phenotypes. Therefore, the effect of Fam49b mutation on T cell development is predominantly due to thymocyte intrinsic functions." This data should be added to the figure.

We thank the Reviewer for the suggestion and have added the (Revised Figure 3—figure supplement 2). Increased expression of CD5 on DP (Revised Figure 3—figure supplement 2A) and a lower frequency of TCRβ^hi^ CD8 SP and TCRβ^hi^ CD4 SP (Revised Figure 3—figure supplement 2B) and enhanced ratio of TCRβ^hi^ CD4 SP and TCRβ^hi^ CD8 SP (Revised Figure 3—figure supplement 2C) were observed in the thymus from Fam49b-KO chimera mice compared to WT chimera mice.

– The data presented in figure 3C and Figure 3 E show a "block" at the transition from TCRbintCD69+ to TCRbhiCD69+ and TCRbintCD5hi to TCRbhiCD5hi, suggesting defective signals in TCRbintCD69+ and TCRbintCD5hi which leads to inefficient transition. Both TCRbintCD69+ and TCRbintCD5hi are still very enriched with DP cells (see supplementary data). So, I wouldn't necessarily conclude that post-positive selection events are affected by this deficiency. This point should be clarified.

We appreciate this important comment. Enhanced TCR-signaling strength intrinsic to Fam49b-KO DP thymocytes may lower the threshold for positive selection, resulting in a diminution in the frequency of thymocytes undergoing death by neglect where positive selection failed. We found that the frequency of thymocytes undergoing death by neglect remained the same in Fam49b-KO thymus, while frequency of thymocytes undergoing clonal deletion was significantly increased (Revised Figure 4A). Therefore, we concluded that “positive selection remains mostly unaffected by lack of Fam49b molecule, while Fam49b plays a more important role in the later stages of T cell development in the thymus”.

– In Figure 4B, the staining of CD69 in CD8^+^ T cells is weird and is suggestive that CD3low cells were not excluded in the gating strategy. Did the authors exclude CD3 low CD8^+^ T cells from their analysis on CD8SP thymocytes in general?

We thank the reviewer for these thoughtful comments. Reviewer #3 also made a similar comment about the issue of ISP subset (Reviewer #3’s comment no. 3.). Previously, we analyzed CD8 SP and CD4 SP without TCRβ pre-gating. As the reviewer pointed out, we pre-gated on TCRβ-positive cells, and excluded TCRβ-negative CD8 and CD4 ISP cells from the conventional CD8 or CD4 SP thymocytes respectively. CD69 expression in CD8 SP resulted in a two peaked histogram (Revised Figure 4B). The expression of CD69 on CD8 SP was comparable between WT and Fam49b-KO mice as before (Revised Figure 4B). We have now clarified TCRβ pre-gating in the corresponding figure legend. We replaced the FACS plots of CD8 SP and CD8 SP in Revised Figure 4B with TCRβ pre-gating.

It is also difficult to conclude that the increased CD5 expression on DP cells is due to higher TCR signaling or rather to a shift in TCR repertoire towards higher affinity TCR. An analysis in TCR transgenic mice would resolve this issue

We appreciate this valuable suggestion. We showed that CD5 expressions were upregulated on DP in Fam49b-KO OT-I mice (Revised Figure 6—figure supplement 1A upper panel). However, the frequency and numbers of CD8 SP were comparable between WT and Fam49b-KO OT-I mice (Revised Figure 6A-6B). We reasoned that the increased TCR signal of DP in Fam49b-KO OT-I mice is still below the threshold for inducing apoptosis in DP and CD8 SP.

– In Figure 4C, the IL-2 production was tested after three days of stimulation. This is too late since the peak of IL-2 production occurs in general 24h after stimulation. Kinetics of IL-2 production will be more informative.

Thank you for your suggestion. It has been reported that accumulation of IL-2 as measured by ELISA steadily increased and then plateaued up to 70 h after in vitro stimulation, even though the peak of IL-2 production occurs in general 24h after stimulation [6]. We speculated that IL-2 levels in supernatant might be increased in 24 h after CD3ε stimulation. And we have removed the data from peripheral total T cell activation (Figure 4C and Supplementary Figure 5) due to qualitative differences between naïve and memory T cells such as cytokine secretion and proliferation. The effect of fam49b on differentiation and activity of peripheral T cells subset is an important question to be addressed in further studies.

– The data presented in Figure 5 concerning the differential impact of Fam49b deficiency on CD8aa TCRab+ IELs, liver iNKT cells and lymph node Tregs Tregs are very descriptive, without any data on the development in the thymus. This should be clarified.

We thank the Reviewer for this suggestion. We found a strong reduction of CD8αα^+^TCRαβ^+^ IELs compared to Treg and iNKT cells. Therefore, we analyzed thymic CD8αα IEL precursors (IELps) in Fam49b-KO mice. It has been reported that there are two distinct CD8αα IELps in the thymus, which are Type A IELps (PD-1^+^) and Type B IELps (PD-1^-^)[1]. We observed a comparable number of Type A IELps, but more Type B IELps in Fam49b-KO mice than in WT mice (Revised Figure 7—figure supplement 3A right panel). Severe CD8αα^+^TCRαβ^+^ IELs deficiency in the gut was at odds with the higher number of thymic IELps in Fam49b-KO mice. By analyzing Fam49b-KO OT-I mice, we found that Fam49b molecules is also required for the maintenance and survival of peripheral T cells (Revised Figure 6). Therefore, it is possible that the lower number of CD8αα^+^TCRαβ^+^ IELs may be the result of impaired maintenance or survival in the periphery, rather than a defect of thymic IELps development. We have included this in the Result section (page 11, lines 262-273).

Reviewer #3 (Recommendations for the authors):1) Figure 1: it would be informative to add the sequencing trace files for the confirmation of the mutation. Furthermore, after which amino acid (what is the protein length of Fam49a and Fam49b) is the stop codon? Is it possible that a truncated form of Fam49a and Fam49b is expressed? Would the antibodies that have been used for the western blots detect truncated forms? Based on the information provided, it is not clear whether the antibodies against Fam94a and Fem49b recognize the N, C terminus or another region of the proteins.

We apologized for the lack of clarity. As the reviewer pointed out, we added the sequencing trace files showing the Fam49a and Fam49b mutation in Revised Figure 1A and 1B below. Fam49a-KO gene ablation is achieved by creating premature stop codon in exon 7, which can be translated into 140 amino acids of truncated Fam49a protein. Fam49b-KO gene ablation is achieved by creating premature stop codon in exon 6, which can be translated into 117 amino acids of truncated Fam49a protein. We clarified this point in the corresponding figure legend. Anti-Fam49a polyclonal antibodies (Σ, SAB 1103179) was produced by synthetic peptide corresponding to amino acids 56-70 of human FAM49a molecules in rabbit. Anti-Fam49b mouse monoclonal antibody (Santa cruz, D-8, sc-390478) binds to 8-20 near the N-terminus of Fam49b of human origin. Therefore, it will be possible that a truncated form of Fam49a and Fam49b protein could be detected by Fam49a or Fam49b antibodies respectively, if the truncated Fam49a or Fam49b proteins are stably expressed in KO mice. However, we did not detect the band expected to be a truncated Fam49a or Fam49b protein by western blotting. Therefore, we believe that a truncated form of Fam49a or Fam49b protein, if expressed, are likely to be expressed very unstably in KO mice.

2) Figure 2A and B: It would be informative to also show the percentages and numbers of TCRb+ cells as well as to pre-gate on TCRb+ cells when displaying CD4 and CD8a expression. Is there an increase in DN cells in the periphery? Which marker was used for B cells? CD19? This is not indicated.

We thank the Reviewer for this suggestion. As the reviewer pointed out, we examined frequencies of CD4^+^ cells and CD8α^+^ cells with pre-gating TCRβ^+^ cells in lymph nodes of WT, Fam49a-KO mice, and Fam49b-KO mice (Author response image 1). Since reduction in the number of CD8^+^ T cells was greater than that of CD4^+^ T cells (Figure 2A), the frequency of CD8α^+^ cells with pre-gating TCRβ^+^ cells decreased in Fam49b-KO mice (Author response image 1). Next, we analyzed the frequency and number of TCRβ^+^CD4^-^CD8α^-^ cell (Double-Negative, DN) T cells in lymph nodes of WT, Fam49a-KO mice, and Fam49b-KO mice. The frequency of DN T cells among total lymphocytes were increased from 0.004% to 0.006% in Fam49b-KO mice (Author response image 1), though the number of DN T cells were comparable (Author response image 1). Fam49b deficiency does not lead to a strong reduction in DN T cells. We used B220 molecules for B cells marker. We clarified this point in the corresponding Figure legend.

**Author response image 1. sa2fig1:** (A) Flow cytometry profiles of the expression of CD4 and CD8 of lymphocytes with pre-gating TCRβ^+^ cells in peripheral lymph nodes from WT, Fam49a-KO, and Fam49b-KO mice. Numbers adjust to outlined areas indicate percentage of cells among total TCRβ^+^ cells lymphocytes. (B) Frequencies of TCRβ^+^CD4^-^CD8α^-^ cell cells among total lymphocytes in lymph nodes from WT, Fam49a-KO mice, and Fam49b-KO mice. Each dot represents an individual mouse. Small horizontal lines indicate the mean of 8 mice. ****p<0.0001 (One-way ANOVA). (C) Numbers of TCRβ^+^CD4^-^CD8α^-^ cell in lymph nodes from WT, Fam49a-KO mice, and Fam49b-KO mice. Each dot represents an individual mouse. Small horizontal lines indicate the mean of 8 mice.

3) In Figure panel 2D: it would be important to pre-gate on TCRb-high cells for determining the CD4/CD8SP ratio. Otherwise, the CD8SP population includes the CD8 ISP subset.

We agree and have incorporated this suggestion throughout our paper. Reviewer #2 also asked a similar question about the issue of ISP subset (Reviewer #2’s comment no. 6.). We previously analyzed CD4 SP and CD8 SP without TCRβ pre-gating. As the reviewer pointed out, we pre-gated on TCRβ-high cells, and excluded TCRβ-low ISP cells from the conventional CD4^+^ SP thymocytes and CD8^+^ SP thymocytes. We observed that frequencies and numbers of TCRβ-high CD4 SP and TCRβ-high CD8 SP were still reduced, and the ratios of TCRβ-high CD4 SP to TCRβ-high CD8 SP were also increased as before. We replaced individual value plot in Revised Figure 2D and described TCRβ pre-gating in the corresponding Figure legend.

4) Figure 3: Did the authors also analyze CD44 and CD62L expression in the mixed BM chimeric mice? This would help to understand whether the altered CD44/CD62L expression (relative reduction of the frequency of naive cells) shown in Figure 2C is T cell-intrinsic or potentially a consequence of (a partial) T cell homeostatic expansion due to T cell lymphopenia.

We have not analyzed CD44 and CD62L expression in chimeric mice. However, we believe that Revised Figure 6 (page 11, lines 262-273) would address this issue because we found new function of Fam49b molecules on peripheral T cells by analyzing Fam49b-KO OT-I mice. Fam49b plays an important role not only in the development of T cells in the thymus but also in the maintenance and survival of peripheral T cells. Therefore, we concluded that reduction of naïve T cells in Fam49b-KO mice was a result of T cell defect in thymus as well as T cell-intrinsic survival defect in the peripheral.

5) The authors conclude that the reduced numbers of SP cells and peripheral T cells is due to a defect in negative selection. How can the authors exclude that the reduced numbers of T cells is not due to a general survival defect of post positive selection SP thymocytes? What about expression of pro-apoptotic and pro-survival factors such as Bim and Bcl2, respectively? And what about IL-7 receptor α expression, which is upregulated during positive selection. The expression of these factors should be analyzed.

To directly address the reviewer’s question, we examined the maturation of SP thymocytes. As pointed out by the Reviewer, SP maturation in the thymus were markedly impaired in Fam49b-KO mice (Revised Figure 5A-5B). Moreover, IL-7Rα expression was partially decreased in semi-mature and mature CD4 SP in Fam49b-KO thymus (Revised Figure 5C). We have described this in the Result section (page 9, lines 194-203). Higher ratio of BIM/BCl^-^2 was detected in naïve Fam49b-KO CD8^+^ OT-I peripheral T cells (Revised Figure 6F-6G). We have described this in the Result section (page 10, lines 225-233).

6) The conclusion that negative selection is altered is based on the results shown in Figure 4A. The frequency of cleaved Caspase-3+ cells is very low. To strengthen the conclusion it would be good to include other approaches that indicate altered negative selection. What happens if WT and Fam49b KO thymocytes are cultured overnight in the presence/absence of anti-CD3 stimulation: would this also reveal changes in cleaved Caspase3?

We agree with the assessment of the Reviewer. The absolute numbers of immature CD8 SP and CD4 SP were decreased from Fam49b-KO mice (Revised Figure 5B), which may support the idea of increased apoptosis in DP thymocytes from Fam49b-KO mice. A more than two-fold increase in apoptosis was observed in Fam49b-KO immature and semi-mature CD4 SP (Revised Figure 5D-5E) and CD8 SP (Revised Figure 5—figure supplement 1A-1B). These results are consistent with our observation that the frequencies of cleaved Caspase-3^+^ cells of signaled thymocytes undergoing clonal deletion were increased in the cortex and medulla.

What about superantigen (SEB)-induced deletion of specific TCR Vb isoforms? This is another readout that can be used to study negative selection.

Per reviewer’s suggestion, we investigated the role of Fam49b in negative selection in a superantigen-mediated deletion model. The C57BL-derived mouse strains express superantigens Mtv-8 and 9, which cause deletion of Vβ3^+^, Vβ5^+^, Vβ11^+^ and Vβ12^+^ TCR clonotypic thymocytes and peripheral T cells when the MHC molecule I–E is also present. Vβ6^+^ thymocytes and peripheral T cells do not recognize Mtv-8 and 9, and were not deleted. To introduce I-E, which is absent on the B6-backcrossed Fam49b-KO mice, we bred them to Balb/c mice which express I-E. In H-2^d/d^ mice, which express I-E molecules. Superantigen-mediated deletion efficiency was comparable between WT and Fam49b-KO mice (Author response image 2). Therefore, we concluded that Fam49b-deficiency does not change negative selection.

**Author response image 2. sa2fig2:** Negative selection in Fam49b-KO mice. B6 mice (H-2b) bearing wild-type or mutated Fam49b allele were backcrossed or not onto Balb/c (H-2d) mice and the expression of indicated Vβ elements in CD4 (left) or CD8 (right) peripheral T cells was analyze3d by flow cytometry.

7) How does altered signaling strength in Fam49b KO thymocytes affect the development of thymic Foxp3+ Tregs and thymic NKT cells? Is there a difference in the frequency of these cells in the thymus? In Figure 5, data are shown from LN (Foxp3) and liver (iNKT) but no data are shown from the thymus.

We have not analyzed thymic Treg and NKT precursors in Fam49b-KO mice because we only observed strong reduction of CD8αα^+^TCRαβ^+^ IELs compared to Treg and iNKT. Therefore, we analyzed thymic CD8αα IEL precursors instead. The results are presents in Revised Figure 7—figure supplement 1A-1C and are described in the Result section (page 11-12, lines 262-273).

8) The data from peripheral T cells are not convincingly supporting the conclusion that TCR signaling strength is altered. Cell proliferation is normal, wouldn't one expect changes in proliferation if signaling strength in KO cells is changed? What about the upregulation of activation markers such as CD69 and CD25? It would also be informative to plot Cell trace violet dilution with intracellular IL-2 staining to determine whether enhanced IL-2 is linked with a particular division stage. What about other cytokines, e.g. TNF or IFNγ? And what happens if cells are stimulated with anti-CD3/CD28 – would there still be a difference in IL-2?

We agree with the Reviewer. We generated Fam49b-KO OT-I transgenic mice and investigated activation of ERK because Fam49b is known to negatively regulate Rac-PAK-ERK axis in T cells [2]. As expected, ERK phosphorylation was markedly increased in Fam49b-KO thymocytes than in WT thymocytes (Revised Figure 6F), which may support the idea of enhanced TCR signaling in Fam49b-KO thymocytes. However, the frequencies and numbers of CD8 SP thymocyte (Revised Figure 6A-6B) as well as peripheral T cells (Revised Figure 6C-6D) were comparable between WT OT-I and Fam49b-KO OT-I mice of 6 weeks of age. We reasoned that the increased TCR signaling of DP in Fam49b-KO OT-I mice still remains below the threshold for inducing apoptosis of DP and CD8 SP in the thymus. And, we removed the data from peripheral T cell activation (Figure 4C and Supplementary Figure 5), since they had little direct relevance to the thesis on the effect of Fam49b on T cell development in the thymus and peripheral survival. We think these changes present improved demonstration for stronger TCR strength in Fam49b-KO mice, and hope that you agree with us.

9) The IEL data are interesting. It would be good to show the gating strategy as well as a representative contour plot for all subsets shown in the diagram in panel 5B and 5C. However, the reduction of the natural IELs could be due to changes in negative selection or – similar to my comment #5 – reduced survival of selected cells (without affecting negative selection). This should be at least discussed as an alternative explanation.

We agree. We have included a new Revised Figure 7-fiure supplement 1 to further illustrate the gating strategy and show representative IELs T cell populations in WT and Fam49b-KO mice. Reviewer #2 also posed a similar question about reduction of the nature IELs. Please see Reviewer #2’s comment number 8 above.

References

1. Ruscher, R., et al., *CD8alphaalpha intraepithelial lymphocytes arise from two main thymic precursors.* Nat Immunol, 2017. 18(7): p. 771-779.

2. Shang, W., et al., *Genome-wide CRISPR screen identifies FAM49B as a key regulator of actin dynamics and T cell activation.* Proc Natl Acad Sci U S A, 2018. 115(17): p. E4051-E4060.

3. Fischer, A.M., et al., *The role of erk1 and erk2 in multiple stages of T cell development.* Immunity, 2005. 23(4): p. 431-43.

4. Eblen, S.T., et al., *Rac-PAK signaling stimulates extracellular signal-regulated kinase (ERK) activation by regulating formation of MEK1-ERK complexes.* Mol Cell Biol, 2002. 22(17): p. 6023-33.

5. D'Souza, W.N., et al., *The Erk2 MAPK regulates CD8 T cell proliferation and survival.* J Immunol, 2008. 181(11): p. 7617-29.

6. Sojka, D.K., et al., *IL-2 secretion by CD4^+^ T cells* in vivo *is rapid, transient, and influenced by TCR-specific competition.* J Immunol, 2004. 172(10): p. 6136-43.

[Editors’ note: what follows is the authors’ response to the second round of review.]

The 3 Reviewers appreciate the important additions contained in the revised manuscript such as the inclusion of data from TCR transgenic mice that improve the quality and impact of the study. However and as outlined below by the 3 Reviewers several precise issues need to be addressed prior to publication in eLife.

We have added many new experimental data per reviewers’ suggestions. We hope that our point-by-point responses to the reviewer’s comments are clear and satisfactory.

Reviewer #1 (Recommendations for the authors):The revised manuscript contains some important additions such as the inclusion of data from TCR transgenic mice that improve the quality and impact of the study. That said, this report is mainly descriptive with very little mechanistic insight. It was hoped that the requested signaling/biochemistry experiments would contribute to mechanism, but as stated below, the results of these studies are not trustworthy. Several issues need to be addressed prior to publication:1. The observation that Fam49a KO mice have no obvious phenotype is not a valid reason for not generating Fam49a/49b double KO (DKO) mice. As mentioned in the review, there could be partial redundancy between Fam49a and 49b and the DKO could have a phenotype more severe than the Fam49b KO. This reviewer does not insist on the generation of the DKO (though it is unfortunate) but does want to make the point that the authors' reasoning for not generating the DKO is flawed.

To response the reviewer's question, we have attempted to make Fam49a/b DKO mice. First, we generate Fam49a-KO Fam49b-Het mice and crossed them with each other. It was expected that 25% of mice born would be Fam49a/b-DKO. However, contrary to our expectations, Fam49a/b-DKO mice could not be obtained in 30 pups aged 3 weeks obtained from three breeding attempts. All pups were either Fam49a-KO Fam49b-WT or Fam49a-KO Fam49b-het. Therefore, we could not proceed with the assay experiment because Fam49a/b-DKO mice were not born. We predicted that loss of the expression of two proteins, Fam49b and Fam49b, induces stillbirth of offspring during pregnancy or early birth. We would like to mention that Rac1 deficient mouse embryo also failed to form appropriate germ cell layers and died at gastrulation [1].

2. The CD69 surface expression results are perplexing. While CD5 surface expression is "analog" and parallels TCR signal intensity, CD69 expression has historically been found to be more "digital" (i.e., CD69+ and CD69- populations rather than a shift in MFI), though the data in Figure 4b indicate an analog pattern (the differences shown are also very slight).

We thank the reviewer for pointing out this important issue and apologize for the confusion caused. Accordingly, we have removed the CD69 expression data in Figure 4B.

3. In Figure 5a and 5b, how do the authors explain normal numbers of semi-mature cells when the numbers of immature and mature cells are reduced?1) normal numbers of semi-mature cells when the number of immature are reduced?

We thank the reviewer for this valuable question. The newly generated SP thymocytes in the thymic medulla undergo further maturation over several days before exit [2]. We believe that compensatory proliferation of semi-mature SP thymocytes populations in the medulla can account for this recovery, even if a small number of immature SP thymocytes migrate from the cortex to the medulla.

2) the numbers of mature cells are reduced when normal numbers of semi-mature cells?

The proper actin cytoskeleton remodeling-dependent signaling is required for maturation of SP thymocyte [3, 4]. Previous study has showed that survival of semi-mature SP thymocytes was also significantly decreased in PAK2-deficient mice [3]. Mechanistically, Rac-Fam49b-PAK2 is required for actin reorganization triggered by TCR in thymocytes [5]. We also observed a twofold increase in cell death in Fam49b-deficient semi-mature SP thymocytes (Figure 5D-E and Figure 5—figure supplement 1A-B). Based on this knowledge and our observation, we suggest that decreased number of mature SP thymocytes from Fam49b-KO mice might have been the result of impaired survival and maintenance of semi-mature SP thymocyte.

4. The results in Figure 5c are not convincing (that there is a difference in the expression of IL-7ra).

In Fam49b-KO mice, IL-7Rα expression was not significantly different in immature and mature SP thymocytes but was only significantly decreased in semi-mature CD4 SP thymocytes (Figure 5C). Consistent with reduced IL-7Rα expression, we also found that apoptosis of semi-mature SP thymocytes was significantly increased in Figure 5D-E. Previous study has demonstrated that Fam49b inhibited TCR signaling via Rac-PAK axis signaling [6]. Moreover, it has been reported that IL-7Rα expression and survival of semi-mature SP thymocytes was also significantly decreased in RAK2-deficient mice [3], suggesting that PAK2 signaling is required for survival of semi-mature SP thymocytes. Therefore, we suggest that Rac-Fam49b-PAK2 axis signaling plays a key role in the survival and maintenance of semi-mature CD4 SP thymocytes in the medulla.

5. In Figure 5d,e how do the authors explain increased cell death of semi-mature cells when these cells were not reduced (Figure 5a,b)?

We reasoned that compensatory proliferation of semi-mature SP thymocytes populations in the medulla can account for this recovery. Like the results of Fam49b-KO mice, there is a paper showing that semi-mature CD4 SP thymocyte survival is reduced in PAK2-deficient mice, although even cell numbers are slightly increased [3].

6. Higher expression of CD5 and CD69 in Figure 6 is not convincing.

We agree with the Reviewer that the increase of CD5 expression on DP thymocyte from Fam49b-KO OT-I mice was not as noticeable (previous Figure 6—figure supplement 1A) compared to the data of DP thymocyte from Fam49b-KO mice in Figure 4B. Moreover, contrary to the predictions, there were no significant differences in frequencies and numbers of CD8 SP thymocytes in Fam49b-KO OT-I mice at 6 weeks (Figure 6A-B), suggesting that Fam49b protein does not have a significant effect on T cell development in the thymus of Fam49b-KO OT-I mice. Thus, we removed the CD5 and CD69 expression of thymus from Fam49b-KO OT-I mice in previous Figure 6—figure supplement 1A-B since they had little direct relevance to the key finding in Fan49b-KO OT-I mice that Fam49b is required for peripheral T cell survival. We think these changes present improved demonstration for impaired survival in Fam49b-KO peripheral OT-I T cells, and hope that you agree with us.

7. Figure 6d, how do the authors explain the low % of naive OTI T cells in the control mice? Typically, >90% of OTI CD8^+^ T cells are naive in lympho-replete mice.

We apologize for the lack of clarity. Percentage of naïve OT-I T cells was calculated as a percentage of naïve CD8 OT-I T cells (CD62L^+^CD44^lo^TCR-Vα2^+^CD8^+^ T cells) among total lymphocytes, not total CD8 OT-I T cells (TCR-Vα2^+^CD8^+^ T cells). About 30% of the total lymphocytes in WT OT-I mice are CD8 OT-I T cells, and about 65% of CD8 OT-I T cells are CD62L^+^CD44^lo^ naïve CD8 OT-I T cells (Author response image 3). Therefore, the proportion of naïve CD8 T cells among all lymphocytes is approximately 20%. As reviewer pointed out, >90% of CD8 OT-I T cells are CD44^lo^ CD8 OT-I naïve T cells (Figure 6D)

**Author response image 3. sa2fig3:** Naïve CD8 OT-I T cells identified by TCR-Vα2^+^, CD8α^+^, TCRβ^+^, CD4^-^, CD62L^+^, and CD44^lo^.

8. Figure 6f,g, these results are the most problematic in the revised paper. pERK should be virtually undetectable in unstimulated thymocytes and T cells. Also, there is no increase in pERK after stimulation indicating that the cells are not activated. Other effectors (pLAT, pZAP, pZAP-70) should be checked. Several experiments should be performed and statistics shown in Figure 6g. Most notably, the authors do not evaluate Rac activation in the mutant mice to show that it is elevated as predicted!1) pERK should be virtually undetectable in unstimulated thymocytes and T cells. Also, there is no increase in pERK after stimulation indicating that the cells are not activated.

We appreciated this important comment. The resting time after thymocyte isolation was increased from 2 hours to 4 hours to prevent the basal expression of pERK before TCR stimulation. We clearly observed that the pERK expression was significantly increased after TCR stimulation. These results are presented in Revised Figure 4C and described this point in the corresponding Materials and methods.

2) Other effectors (pLAT, pZAP, pZAP-70) should be checked.

As the reviewer pointed out, we examined the activation of key TCR signaling cascade component (ZAP-70, LAT, PLCγ1, and ERK). Fam49b deficiency led to prolonged increases in all the downstream phosphorylation events tested (a new Revised Figure 4C), suggesting that Fam49b-deficiency thymocytes received enhanced TCR signaling.

3) Several experiments should be performed and statistics shown in Figure 6g.

We apologize for the lack of clarity. We analyzed the expression of BIM and BCl^-^2 molecules in naïve T cells and total thymocytes by western blot from three independent experiments. We observed that BIM/BCl^-^2 ratio was markedly increased in Fam49-KO thymocytes and naïve T cells. These results are presented in new Revised Figure6—figure supplement 2A-2B and described this point in the corresponding Figure legend.

4) the authors do not evaluate Rac activation in the mutant mice to show that it is elevated as predicted!

Fam49b has been identified as an inhibitor of TCR signaling via Rac-PAK axis in Fam49b-KO Jurkat T cells [6]. Consistent with the results of this paper, PAK phosphorylation was also dramatically elevated in Fam49b-KO thymocytes (a new Revised Figure 4D). Therefore, we concluded that Fam49b-deficiency thymocytes may receive enhanced TCR signaling via Rac-PAK aixs.

9. The results in Figure 7 are interesting but the authors did not perform the requested experiment, namely to generate CD28 KO / Fam49b KO (DKO) mice to test if negative selection is indeed increased in Fam49b mice. Deletion of CD28 would allow the rescue of otherwise negatively selected thymocytes, and an increase in DN, TCR+, PD-1+ cells in CD28/Fam49b DKO mice compared to CD28 KO mice would provide strong support for their hypothesis that negative selection is increased in the absence of Fam49b.1) The results in Figure 7 are interesting but the authors did not perform the requested experiment, namely to generate CD28 KO / Fam49b KO (DKO) mice to test if negative selection is indeed increased in Fam49b mice

The intensity and duration of TCR signaling based on TCR affinity to self-peptide:self-MHC complex are the major determinants of positive and negative selection [7]. Negative selection, also known as clonal deletion, involves inducing apoptosis in thymocytes that bind with high affinity to self-peptide:self-MHC complex during the DP and SP stages, while positive selection induce survival and differentiation programs in DP thymocytes. We found that enhanced TCR-signaling strength intrinsic to Fam49b-KO DP thymocytes (Revised Figure 4C) led to excessive clonal deletion in the cortex and medulla (Figure 4A), while the frequency of thymocytes undergoing death by neglect remained the same in Fam49b-KO thymus (Figure 4A).

How might enhanced Rac activity lead to the enhanced clonal deletion, involving apoptosis, in Fam49b-KO mice? Rac is known to regulate actin reorganization in thymocyte and T cells [5]. Negatively regulated Rac-driven cytoskeleton remodeling of Fam49b-deficient thymocytes could attenuate protrusion and chemoattractant-induced cell migration process because Fam49b-deficient cells showed increased cellular spread and reduced protrusion-retraction dynamics [8, 9]. Moreover, it has been reported that negative selection occurs via lengthy interactions between T cells and APCs, whereas positive selection are transient interactions [10]. Therefore, it is possible that altered cytoskeleton remodeling activity in Fam49b-deficient thymocytes contributed to their elevated TCR-signaling strength and enhanced clonal deletion, perhaps by prolonging interactions with thymic APCs. We have described this possibility in the Discussion section (page 14, lines 315-325)

2) Deletion of CD28 would allow the rescue of otherwise negatively selected thymocytes, and an increase in DN, TCR+, PD-1+ cells in CD28/Fam49b DKO mice compared to CD28 KO mice would provide strong support for their hypothesis that negative selection is increased in the absence of Fam49b.

As reviewer suggested, it has been reported that TCRαβ^+^ DN IEL precursor (IELp) would arise from autoreactive cells diverted from clonal deletion in response to strong TCR signaling [11, 12]. CD28–deficient mice have more TCRαβ^+^ DN IELp in the thymus, suggesting that CD28-mediated costimulation is needed to induce autoreactive thymocytes to undergo clonal deletion that have received strong TCR signaling during negative selection [11]. Based on our observation of the dramatic loss of CD8αα+TCRαβ+ IELs in Fam49b-KO mice (Figure 7A), we postulated that TCRαβ^+^ DN IEL precursor might be decrease in thymus. Contrary to our expectations, TCRαβ^+^ DN IELp was slightly increased (Author response image 4).

How might enhanced Rac activity lead to the increase of thymic IELp in Fam49b-KO mice? While both negative selection and agonist selection of IELp are directed by a strong TCR signal, the factors that specify these divergent fates are complicated and remain unclear [13]. Interestingly, thymocytes undergoing agonist selection into IELp exhibited a rapid and confined migration pattern, in contrast to negatively selecting cells, which showed arrested migration [13]. Fam49b-deficient cells showed increased cellular mobility [9]. It is tempting to speculate that overactivation of Rac-1 in Fam49b-KO mice might rescue IEL precursors from negative selection, perhaps by favoring confined migration over migratory arrest after encountering with agonist ligands. We have described this possibility in the Discussion section (page 14-15, lines 339-350)

**Author response image 4. sa2fig4:** Numbers of TCRβ^+^ DN IEL precursor cells in WT and Fam49b-KO mice at 6-7 weeks of age. Each dot represents an individual mouse. Small horizontal lines indicate the mean of 6-7 mice. **p=0.0031 (Mann-Whitney test). Data are representative of three experiments.

3) The results in Figure 7 are interesting but the authors did not perform the requested experiment, namely to generate CD28 KO / Fam49b KO (DKO) mice to test if negative selection is indeed increased in Fam49b mice

It has been reported that CD8αα^+^TCRαβ^+^ IELs develop from two main thymic IELp, which are PD-1^+^(T-bet^-^) IELp and PD-1^-^(T-bet^+^) IELp [12]. Consistent with the proposal of clonal deletion as an alternative fate for IELp, the number of PD-1^+^ IELp and PD-1^-^ IELp was greater in the thymus of mice deficient in the proapoptotic protein Bim (Bim-deficient mice; a model in which normally deleted T cells are ‘rescued’ [14]) than in that of WT mice. In addition, CD28-KO mice, in which self-reactive thymocytes are diverted into the CD8αα IEL lineage [11], had only more PD-1^+^ IELp cells, not more PD-1^-^ IELp, than WT mice [12]. These characteristics suggested that the PD-1^+^ IELp and PD-1^−^ IELp are represent separate lineages, and CD28 costimulatory signaling would be only involved in the generation of PD-1^+^ IELp. We have examined the frequencies and numbers of these two IELps (PD-1^+^ IELp and PD-1^-^IELp) in the thymus from WT and Fam49b-KO mice at 6-7 weeks of age. Interestingly, we observed comparable numbers of PD-1^+^ IELp, but more PD-1^-^ IELp in Fam49b-KO mice than in WT mice (Figure 7—figure supplement 3A right panel). We reasoned that Fam49b might be specifically involved in development of thymic PD-1^-^ IELp lineage, but not PD-1^+^ IELp lineage, in a CD28-independent manner. In addition to the complexity of thymic IELps, various factors could regulate the generation of agonist selected thymic IELps [13]. Therefore, we think that CD28/Fam49b DKO mice could not be a good animal model to test whether negative selection is increased. The effect of fam49b on generation of thymic IELp is an important question to be addressed in further studies. We hope the above explanation sufficiently answers your questions, and we hope you agree with us.

10. Supplement 3a and 3b are mentioned in the text but not included in the figures.

We thank the reviewer for noticing and have included Figure 7—figure supplement 3A-3B in the figures.

Reviewer #2 (Recommendations for the authors):We would like to thank the authors for performing several experiments that resolved some issues raised during my first reviewing process. These new experimental data sets revealed that Fam49b, besides its impact on thymic selection, plays an important role in T cell survival in the periphery. Based on these new findings, the authors should amend the discussion that is mainly focused on thymic selection.

We thank the Reviewer for the suggestion and have amended the Discussion section (page 14, lines 327-337). We have also revised the title and the Introduction section (page 5, lines 89-91) to emphasize that Fam49b protein plays an important role in peripheral T cell survival as well as thymocytes.

In addition, the data presented concerning the impact Fam49b on TCR signaling are not convincing and concerns only ERK without quantification and statistical analyses. This part should be strengthened by analyzing other signaling pathways.

Fam49b has been identified as an inhibitor of TCR signaling via Rac-PAK axis in Fam49b-KO Jurkat T cells [6]. Therefore, we investigated the activation of key TCR signaling cascade component (ZAP-70, LAT, PLCγ1, and ERK). We observed that Fam49b deficiency led to prolonged increases in all the downstream phosphorylation events investigated in thymocytes (a new Revised Figure 4C). Moreover, PAK phosphorylation was also dramatically elevated in Fam49b-KO thymocytes (a new Revised Figure 4D). Therefore, we concluded that Fam49b-deficiency thymocytes received enhanced TCR signaling via Rac-PAK aixs.

[Editors’ note: what follows is the authors’ response to the third round of review.]

The manuscript has been improved but there are some remaining issues that need to be addressed, as outlined below:The authors' explanation for the perplexing results in Figure 5a and 5b (i.e., that proliferation differences could explain why the numbers of semi-mature cells are normal in the KO when the numbers of immature and mature cells are reduced) is unsatisfactory. The reviewer is unaware of any data showing that SP thymocytes proliferate at any stage of their maturation. If I am mistaken, please provide the citation for these findings. I think it more likely that the gating (particularly distinguishing CD62L-lo from -neg and CD69-lo from -neg) is prone to error. In any event, the finding that semi-mature cells are not reduced but immature and mature cells are reduced it very confusing. Because I believe this is a technical problem, I suggest just showing the results for only immature and mature cells to avoid this apparent inconsistency.

We thank the reviewer for pointing out this important issue and apologize for the confusion caused. Accordingly, we have removed the CD69 expression data in Figure 4B.

1) Legend says plots are LN T cells not total LN cells, but Figure 2c shows a large population of CD62L-CD44- cells. Could it be that the cells in Figure 2c are total LN cells?

The plots are LN T cells not total LN cells.

2) In Figure 2c, what are CD62L-CD44- cells? To our knowledge, no CD62L-CD44- T cell populations have been previously described.

According to the Biolegend's Naïve/Nemory T cell analysis kit manual, CD62L^-^CD44^low^ T cells (named effector T cells) exist in mice (https://www.biolegend.com/en-gb/products/mouse-naive-memory-t-cell-id-panel-9757). These CD62L^-^CD44^low^ T cell population are also observed in other paper (PLZF induces the spontaneous acquisition of memory/effector functions in T cells independently of NKT cell-related signals J Immunol. 2010 Jun 15;184(12):6746-55(PMID: 20495068)). However, characteristics of these T cells has been rarely studied. According to a paper published in the journal of PNAS in 2021, these T cells is important for age-related resistance to PD-1 blockade [1].

Line 163 "Taken together, these results suggest that positive selection remains mostly unaffected by the lack of Fam49b molecule, Fam49b plays a more important role in the later stages of T cell development in the thymus." What "later stages" of development are they referring to? Do they mean survival? Also, the reduction of TCR-hi CD69+ cells (Figure 3c) suggests that late stages of positive selection are affected. I would maintain that "post-selection" means CD69-neg and that any CD69+ cell is undergoing selection.1) What "later stages" of development are they referring to? Do they mean survival?

We use “later stages” to refer to post-positive selection (TCRβ^hi^CD69^+^) and the mature thymocytes (TCRβ^hi^CD69^-^). Lower frequency of post-positive selection (TCRβ^hi^CD69^+^) and the mature thymocytes (TCRβ^hi^CD69^-^) may reflect survival issues following positive selection.

2) Also, the reduction of TCR-hi CD69+ cells (Figure 3c) suggests that late stages of positive selection are affected. I would maintain that "post-selection" means CD69-neg and that any CD69+ cell is undergoing selection.

It is possible that strong TCR signaling due to Fam49b protein deficiency may allow Fam49b-KO thymocytes that would otherwise die by "death by neglect" to pass "positive selection". We thought that if this possibility occurred in Fam49b-KO mice, the frequency of thymocytes dying by “death by neglect” would be decreased. However, we observed that the frequencies of cells to be eliminated through “death by neglect” (cleaved-caspase 3^+^TCRβ^-^CD5^-^ cells) were similar between Fam49b-KO and WT mice (Figure 4A bottom row). Thus, we described that “positive selection remains mostly unaffected by the lack of Fam49b molecule”.

Since the reader encounter Figure 4A after Figure 3C, the statement “positive selection remains mostly unaffected by the lack of Fam49b molecule” may be unnecessarily confusing to reader. Moreover, as the reviewer points out, we cannot completely dismiss the possibility that the lower frequency of TCRβ^hi^CD69^+^ thymocytes in Figure 3C may indicate impairment in the later stages of positive selection. Therefore, we have modified sentences as follows “Taken together, these results suggest that Fam49b play an important role of T cells development, especially in TCRβ^hi^CD69^+^ and TCRβ^hi^CD69^-^ thymocytes”

Line 126 "we concluded that Fam49a is unlikely to 127 play a significant role in T cell development" is obviously misleading since the authors mention in their response that the double Fam49a/49b KO has a more severe phenotype than the Fam49a KO.

As suggested by the reviewer, and considering the possibility that the Fam49a protein could affect T cell development, we have removed the sentence "we concluded that Fam49a is unlikely to play a significant role in T cell development" from lines 126-127.

Since the authors did make OTI TCR transgenic Fam49b KO mice then it would be very helpful to see analysis of positive selection (e.g., as shown in Figure 3c,e, Figure 4b, Figure 5b) to show if the affects observed with polyclonal mice are observed with TCR transgenic mice.1) analysis of positive selection (e.g., as shown in Figure 3c,e, Figure 4b)

One limitation of OT-I transgenic mice is that the transgenic TCR is highly expressed throughout the development of thymocytes [2]. In polyclonal C57BL/6 WT mice, αβTCR, CD4, and CD8α are expressed at the DP thymocyte stage, resulting in positive and negative selection. However, in WT and Fam49b-KO OT-I mice, αβTCR expression significantly increases from the DN stage (Author response image 5). As a result, identifying populations based on TCRβ and CD69/CD5 expression to further characterize positive selection is challenging.

**Author response image 5. sa2fig5:** (A) Flow cytometry analyzing the expression of CD4 and CD8α in thymocytes from WT and Fam49b-KO OT-I mice (upper). Flow cytometry analyzing the expression of OT-I-specific variable region TCR-Vα2 and TCRβ on gated CD4^-^CD8α^-^ DN thymocytes from WT and Fam49b-KO OT-I mice at 6 weeks of age (lower). Shown are representative data of three-four mice per genotype.

However, CD5 expression do correlate with the strength of TCR signaling. In polyclonal C57BL/6 WT mice, most DP thymocytes do not undergo positive or negative selection, which requires TCR engagement of pMHC, and thus do not upregulate CD5 expression (Author response image 6). WT and Fam49b-KO OT-I mice have a large population of TCR-Vα2^+^ DP thymocytes that undergo positive selection, as indicated by an increase in the CD5^+^ population compared to polyclonal WT mice (Author response image 6). An interesting observation is that Fan49b-KO OT-I mice express high level of CD5 on TCR-Vα2^+^ DP compared to WT OT-I mice, which is consistent with the phenotype of polyclonal Fam49b-KO mice (shown in Figure 4B). This observation indicates that Fam49b-KO OT-I DP thymocytes had received stronger TCR signaling than the WT OT-I DP thymocytes. We postulated that enhanced TCR-signal strength of Fam49b-KO OT-I DP thymocytes could be diverted into negative selection from positive selection. However, contrary to the predictions, there were no significant differences in frequencies and numbers of CD8 SP thymocytes between WT and Fam49b-KO OT-I mice (Figure 6A and Figure 6B). Therefore, we suggest that fam49b deficiency did not induce strong TCR signaling enough to pass the threshold for the negative selection in Fan49b-KO OT-I mice.

**Author response image 6. sa2fig6:** (A) Expression of activation marker CD5 on TCR-Vα2^+^ DP and TCR-Vα2^+^ CD8 SP thymocytes from WT and Fam49b-KO OT-I mice at 6 weeks of age. (B) Geometric MFI of CD5 on TCR-Vα2^+^ DP. Small horizontal lines indicate the mean of 4 mice. *p=0.0278 (Mann-Whitney test). Shown are representative data of three independent experiment.

When analyzing thymocytes using CD69 and TCRβ expression, compared to polyclonal C57BL/6 WT mice (Figure 3C), most of the TCR-Va2^+^ thymocytes in WT and Fam49b-KO OT-I mice are in Stage 4 (TCRβ^hi^CD69^-^) (Author response image 7). No significant difference was observed between WT and Fam49b-KO OT-I mice (Author response image 4).

When analyzing thymocytes using CD5 and TCRβ expression, compared to polyclonal C57BL/6 WT mice (Figure 3E), Stage 3 (TCRβ^int^CD5^hi^) and Stage 4 (TCRβ^hi^CD5^hi^) account for the largest proportion of TCR-Va2+ thymocytes in WT and Fam49b-KO OT-I mice (Author response image 5). No significant difference was observed between WT and Fam49b-KO OT-I mice (Author response image 7).

**Author response image 7. sa2fig7:** (A) Differential surface expression of CD69 and TCRβ was used to identify thymocyte population of different maturity in WT and Fam49b-KO OT-I mice. (right) Dot Plots show percentages of different thymocyte subpopulations among TCR-Vα2 positive thymocytes from WT and Fam49b-KO OT-I mice. Data are representative of two experiments. (B) Differential surface expression of CD5 and TCRβ was used to identify thymocyte population of different maturity in WT and Fam49b-KO OT-I mice. (right) Dot Plots show percentages of different thymocyte subpopulations among TCR-Vα2 positive thymocytes from WT and Fam49b-KO OT-I mice. Data are representative of two experiments.

2) analysis CD8SP maturity (e.g., as shown in Figure 5b)

We observed that CD5 expression levels were similar between WT and Fam49b-KO OT-I CD8 SP thymocytes (Author response image 5). When analyzing the TCR-Va2^+^ CD8 SP thymocytes of WT and Fam49b-KO OT-I mice using the expression of CD69 and CD62L, compared to polyclonal C57BL/6 WT mice (Figure 5E), the CD69^+^CD62L^-^ population was significantly reduced in WT and Fanm49b-KO OT-I mice (Author response image 8). Additionally, maturity of CD8 SP thymocytes was similar between WT and Fam49b-KO OT-I mice (Author response image 8).

**Author response image 8. sa2fig8:** (A) Frequencies of immature (CD62L^lo^CD69^hi^), semi-mature (CD62^lo^CD69^lo^), and mature (CD62^hi^CD69^lo^) in the TCR-Vα2^+^TCRβ^+^ CD8 SP thymocytes from WT and Fam49b-KO OT-I mice. Numbers adjust to outlined areas indicate percentage of each population among total TCR-Vα2^+^TCRβ^+^ CD8 SP thymocytes. Shown are representative data of three-four mice per genotype. (B) Quantification of cell numbers of immature and mature fraction in TCR-Vα2^+^TCRβ^+^ CD8 SP thymocytes. Each dot represents an individual mouse. Data are representative of two experiments.

1. Nakajima, Y., et al., *Critical role of the CD44(low)CD62L(low) CD8(+) T cell subset in restoring antitumor immunity in aged mice.* Proc Natl Acad Sci U S A, 2021. 118(23).

2. Hu, Q., et al., *Examination of thymic positive and negative selection by flow cytometry.* J Vis

[Editors’ note: what follows is the authors’ response to the fourth round of review.]

The manuscript has been improved but there is a remaining issue that need to be addressed, as outlined below:Therefore, prior to accepting the manuscript please remove the data and the text corresponding to Figure 2c as requested below by Reviewer #1.Reviewer #1 (Recommendations for the authors):I appreciate the authors efforts to respond to my concerns/comments. I accept their response to all of the points except for the problem with Figure 2c. It is true that there is a very small percentage of CD44-CD62L- T cells that can be detected in B6 mice, however this percentage is typically <5% of CD4 and CD8 cells. In Figure 2c, 16% of B6 CD4^+^ T cells are CD62L-CD44- and 30% of B6 CD8^+^ T cells are CD62L-CD44-. This is abnormal. Further, CD62L+CD44+ central memory cells are missing from the B6 CD4 subset in Figure 2c but are present in B6 CD8^+^ cells. This is also abnormal. Altogether this means that there is a problem with the FACS experiment used for Figure 2c.

We have designed a new multicolor flow cytometry panel to minimize overlap between fluorescence signals. We identified a lower frequency of CD44^lo^CD62L^-^ T cell subset than before (10% of CD4 T cells and 13% of CD8 T cells), but still a higher percentage than the reviewer stated (Author response image 9). However, importantly, the number of naïve (CD44^lo^ CD62L^+^) CD4^+^ and CD8^+^ T cells were still significantly reduced in Fam49b-KO mice compared to WT mice. Approximately 5-7% of CD62L^+^CD44^+^ CD4 T cells were identified in WT mice.

If the authors are unable to obtain believable CD44 vs CD62L FACS profiles for their control mice then the results from KO mice cannot be interpreted and Figure 2c should be removed as it is misleading as is.

Since the frequency of CD44^lo^CD62L^-^ T cell subset remains higher than what the reviewer pointed out, we have, in accordance with the reviewer’s suggestion, removed Figure 3C and related text in the revised manuscript. We believe that this revision will help minimize any potential confusion for reader without affecting the overall argument of the manuscript.

**Author response image 9. sa2fig9:** Expression of CD44 and CD62L on T cells (left) and absolute number of T cell subset (right) in peripheral lymph nodes in CD4 T cells (upper) and CD8 T cells (lower) from WT and Fam49b-KO mice. T subset with phenotype of naïve (CD62L^+^CD44^lo^), acute effector (CD62L^-^CD44^lo^), effector memory (CD62L^-^CD44^hi^), and central memory (CD62L^+^CD44^hi^). Numbers adjust to outlined areas indicate percentage of T cells subset among total T cells. Each dot represents an individual mouse. Small horizontal lines indicate the mean of 5-6 mice.